# Systematic Integration of Energy-Optimal Buildings With District Networks

**Raluca Suciu \*, Paul Stadler, Ivan Kantor 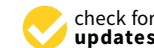, Luc Girardin and François Maréchal**

Industrial Process and Energy Systems Engineering (IPESE), École Polytechnique Fédérale de Lausanne, CH-1951 Sion, Switzerland

**\*** Correspondence: raluca-ancuta.suciu@epfl.ch

**Abstract:** The residential sector accounts for a large share of worldwide energy consumption, yet is difficult to characterise, since consumption profiles depend on several factors from geographical location to individual building occupant behaviour. Given this difficulty, the fact that energy used in this sector is primarily derived from fossil fuels and the latest energy policies around the world (e.g., Europe 20-20-20), a method able to systematically integrate multi-energy networks and low carbon resources in urban systems is clearly required. This work proposes such a method, which uses process integration techniques and mixed integer linear programming to optimise energy systems at both the individual building and district levels. Parametric optimisation is applied as a systematic way to generate interesting solutions for all budgets (i.e., investment cost limits) and two approaches to temporal data treatment are evaluated: monthly average and hourly typical day resolution. The city center of Geneva is used as a first case study to compare the time resolutions and results highlight that implicit peak shaving occurs when data are reduced to monthly averages. Consequently, solutions reveal lower operating costs and higher self-sufficiency scenarios compared to using a finer resolution but with similar relative cost contributions. Therefore, monthly resolution is used for the second case study, the whole canton of Geneva, in the interest of reducing the data processing and computation time as a primary objective of the study is to discover the main cost contributors. The canton is used as a case study to analyse the penetration of low temperature, $CO_2$-based, advanced fourth generation district energy networks with population density. The results reveal that only areas with a piping cost lower than 21.5 k€/100 $m^2_{ERA}$ connect to the low-temperature network in the intermediate scenarios, while all areas must connect to achieve the minimum operating cost result. Parallel coordinates are employed to better visualise the key performance indicators at canton and commune level together with the breakdown of energy (electricity and natural gas) imports/exports and investment cost to highlight the main contributors.

**Keywords:** optimal cities; energy autonomy; low-carbon resources; multi-energy networks; parametric optimisation; $CO_2$ networks

## 1. Introduction

Increasing population, urbanization and rapid industrialization corresponds to parallel and continuous increases in world energy demand, where up to 65% of the energy consumption comes from urban areas [1]. While the consumption of major sectors, such as commercial, industrial, transportation and agriculture are relatively well-understood due to their centralized ownership, self-interest in reducing the energy consumption and high level of regulation, the residential sector is an energy sink which is difficult to characterize, since it encloses a large variety of geometries, structure sizes and envelope materials. At the same time, privacy concerns restrict energy consumption data collection and distribution and detailed metering of households bears high costs. Nevertheless, Pachauri et al. reports

that there is a great potential to achieve significant reductions in energy consumption, mainly in the building sector, at a relatively modest cost [2], which highlights the requirement to better understand the defining characteristics of energy consumption in this sector.

Major end-use energy consumption groups in the residential sector are: space heating and cooling, energy required to overcome thermal flows through the building envelope, by conduction, radiation and through air infiltration/ventilation; domestic hot water-energy consumed to heat water to the comfort temperature; appliances and lighting-energy needed to operate appliances (e.g., refrigerator, electronics) and for supplying appropriate lighting. Fossil fuels are currently the main energy sources to supply these demands [3]; however, they have a high environmental impact and limited reserves which also correspond to fluctuating prices, which affects national economies and results in a prominent interest in using renewable energy sources. Renewable energy comes from a variety of sources, such as biomass, geothermal heat, ocean waves, sun, tides, water and wind. Hybrid (i.e., multi-source) renewable energy systems are favored over single sources since they are more reliable, more efficient, require less energy storage capacity and have lower levelized life cycle electricity generation cost under optimum design [4]. Multi-source generation makes hybrid system solutions complex, thus a techno-economic analysis of these systems is essential to ensure the optimal use of renewable sources. This, in turn, requires models and software which can be employed for design, optimization and techno-economic planning.

Another dilemma that arises with integration of renewable energies is the mismatch between renewable energy supply and demand profiles in the residential sector, which is often pronounced and requires extensive storage solutions [5]. Heat storage solutions already exist at small scale in individual buildings and via district heating networks (DHNs) in large bore-hole storage systems. Alternative solutions exist for multi-energy systems, such as power-to-gas, fuel cells, electric/hydrogen mobility and large scale batteries [6,7].

Balancing energy demand and supply both spatially and temporally can be modeled using computational methods, such as mathematical programming, among which linear programming techniques have been used to optimize multi-energy systems for more than thirty years [8]. Generally, there is a separation of topics in residential energy system analysis based on the scale, namely: individual building scale and urban scale. The former focuses principally or solely on the building itself and omits any relationship with the urban environment. It treats a building as an independent object, isolated from the built environment; however, real buildings are connected to their surroundings through physical means (infrastructure) and users (residents, workers). The latter scale focuses on the entire system, often without details at the building scale. Therefore, there are improvements to be made by coupling building-level models with those at the urban level while also using detailed equipment models (e.g., energy conversion technologies, heat pumps (HP)). Linking buildings with district systems requires tools for design, sizing, operation and control of energy system components, buildings and district networks. An even larger challenge, though, is to provide simple tools, which can aid decision-makers at an early stage in the design process at both the building and urban levels.

This paper proposes a double-optimisation approach with meta-models [9,10] for the design and optimisation of urban systems at building and urban levels, with interaction between the two scales, including renewable energy integration and long-term energy storage solutions. The connection between the building and the urban level is realised through a low-temperature $CO_2$ district energy network and meta-models models are used to integrate building solutions into the district optimisation. Therefore, this paper contributes a novel approach for optimal design of urban energy systems, coupling optimal solutions for individual buildings with the larger energy system to provide guidance for holistic urban energy system design. Additionally, this work provides unique insights into various objectives of such systems and the inherent balance between them, providing a set of optimal solutions to be ultimately selected by decision-makers. Section 2 reviews the main tools and approaches currently employed for this purpose and their limitations, Section 3 presents the mathematical formulation and the case studies considered, Section 4 shows the results and conclusions are drawn in Section 5.

## 2. State of the Art

Energy use in the residential sector has been studied extensively, across a variety of fields, such as civil engineering, architecture, economy, environmental assessment, sociology, transport, city and regional planning. Energy consumed in this sector is generally classified as either embodied or operational. Embodied energy is the energy required to produce and transport materials to the construction site and for the construction process itself, while operational energy is consumed for the daily use of the building to provide electricity, water, hot water, ventilation, heating and cooling.

A clear distinction in the scale of the analysis arises when trying to summarise the research in the area, namely at the individual building and urban scales [1]. Research at the individual building scale usually covers topics such as building materials used, architectural design, structural and operational system and construction. Developments in the area include improving the accuracy of the models and reducing the computation time of the assessment [11], analysing the results with different objectives [12] techniques to reduce energy and $CO_2$ emissions. Kofoworola et al. showed a combination of energy savings measures to reduce the electricity consumption in a typical office building in Thailand by 40–50% [13]. Ochoa et al. stated that the usage phase of buildings accounts for the largest share of the energy use and environmental impact, followed by the construction phase, while the disposal phase is negligible from both perspectives [14]. Junilla et al. presented the elements in the life-cycle assessment of office buildings which cause the highest emissions and should therefore be targeted for improvement [15] and in a similar study concluded that lighting, HVAC systems and outlets, manufacturing and maintenance of steel, manufacturing of concrete and paint and water use have the largest environmental impacts in office buildings [16].

The second scale of analysis for energy use in the residential sector is the urban scale. Research at this scale typically covers topics such as urban form, density, transportation, infrastructure and consumption. Studies in the field focus mainly on quantification of energy use, transportation infrastructure, water infrastructure, construction, and modeling of energy use in urban systems. Glaeser and Kahn studied the energy use and environmental impact due to driving, public transit, providing heating and electricity in households and found a strong negative correlation between emissions and land use regulations, leading them to conclude that cities have significantly lower emissions compared to suburban areas[17]. Kennedy et al. performed a study on ten global cities, showing correlations between public transit quality and personal income, and between heating and industrial fuel use [18]. Troy et al. quantified the embodied energy in urban areas and found it to be more significant than previously supposed and suggested that knowing the embodied energy consumed can be used for control tool development [19].

Jones et al. assessed energy consumption and environmental impact in urban areas due to transportation, energy, water, waste, food, goods and services, and suggested that results were highly dependent on the basic demographic characteristics of the area studied [20]. Regarding energy use modeling, Howard et al. developed a model to estimate end-use energy intensity in New York, as a tool for cost-efficient policies regarding renewable energy efficient solutions [21]. Gurney et al. used simulation tools, traffic data, power production reporting and local air pollution reporting to build a model which quantified $CO_2$ emissions across the city of Indianapolis [22]. Keirstead et al. reviewed approximately 220 papers on urban energy system modeling and concluded that the four most common challenges are data quality and uncertainty, model integration, model complexity and policy relevance [23]. They also concluded that urban energy system models have a significant potential of moving toward a more integrated perspective, which could capture their intricacies.

While these references offer a first insight into multi-scale integration analysis, additional methodological developments are required to directly address the interaction between scales. In view of that, this paper proposes a method which combines the work of Stadler et al. [24] on building optimisation at building level with the work of Suciu et al. [25] on optimisation at district level, to perform a detailed multi-level energy integration optimisation. The link between the building

and the urban level is realised through a low-temperature district heating and cooling network and meta-models are employed to embed the building solutions into the district level optimisation.

*Low Temperature DHC Networks*

Low temperature district energy networks (DENs) provide a low temperature source, which can be used for heating via decentralized heat pumps, directly for cooling, indirectly as a low temperature source for chillers and can recover waste heat from processes and other buildings in the proximity; they are also often linked to large seasonal storage in the form of borehole fields [26].

Low-temperature networks have been discussed in the literature, for example, De Carli et al. performed an energo-economic analysis of a small-scale, low-temperature district heating and cooling network in Italy [27], Bestenlehner compared a low-temperature and a conventional district heating network in a quarter of Stuttgart [28], Ruesch modeled the time evolution of large borehole fields connected to low temperature district heating networks [26], Kräuchi et al. modelled a low-temperature district heating and cooling network using the IDA indoor climate and energy (IDA ICE) simulation software [29] and Molyneaux et al. performed an enviro-economic optimisation for low-temperature heat networks with heat pumps [30].

This work analyzes both conventional networks and low temperature refrigerant ($CO_2$)-based networks. Weber and Favrat introduced the idea of distributing $CO_2$ in the district energy networks at a temperature below the critical pressure of 74 bar. $CO_2$ networks (Figure 1) use a double-pipe system to deliver both heating and cooling services. A pressure of 50 bar is suggested for use in the network to remain within the saturation temperature range of 12–18 °C, which allows network operations to leverage the latent heat and small pressure difference between liquid and gas phases to provide cooling services by gas expansion. Unlike water-based networks currently in place in several cities, $CO_2$ networks use phase change to realize the heat transfer and allow cooling services to provide heating, which is not possible with conventional systems. The approach is based on a $CO_2$ "closed-loop" concept, i.e., except for leaking (considered negligible) no $CO_2$ enters/leaves the network.

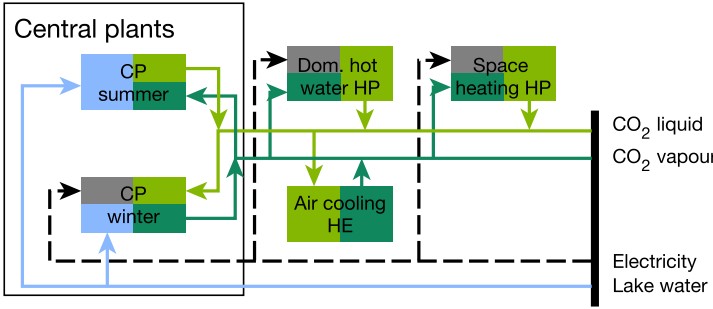

**Figure 1.** $CO_2$ network schematic representation.

$CO_2$ networks have also been integrated with advanced technologies for energy storage and heat integration, such as power-to-gas [25]. Power-to-gas systems use electricity in periods of high production (summer) to produce hydrogen and oxygen by water electrolysis and then methane in a Sabatier reaction, which is stored to provide electricity and heat during cold periods or periods of low electricity production (Figure 2). The waste heat of the co-generation system is first used in a steam network to produce electricity with the remaining low temperature heat used to vaporize $CO_2$, which is used to provide heating services.

This work proposes a method which links analysis and optimisation in individual buildings with urban-level systems through low temperature $CO_2$ networks and long term power-to-gas storage systems. The method proposed uses a double optimisation approach with surrogate models, using two different time scales: monthly averages and typical days.

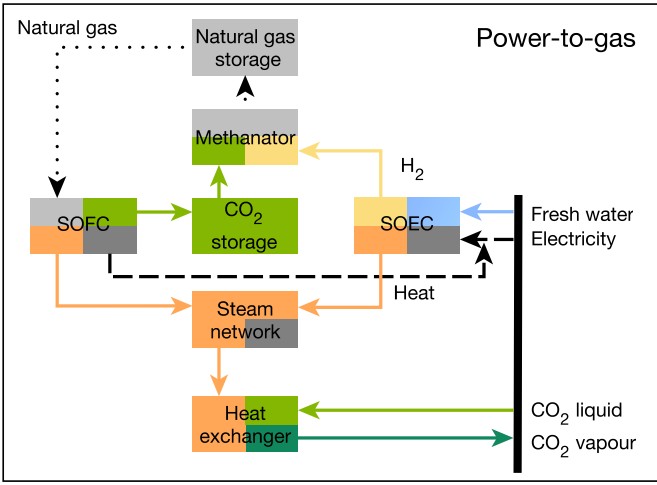

**Figure 2.** P2G schematic representation.

## 3. Materials and Methods

### 3.1. Methodology Overview

The proposed method models energy systems using a double optimisation approach with meta-models (Figure 3). The first optimization is performed at the building level, where different utilities can be chosen, such as photo-voltaic (PV) panels with short-term electricity storage (batteries), $CO_2$ and air-water heat pumps, co-generation units, heat storage tanks, domestic hot water tanks, heat exchangers (HEs) for cooling and electrical heaters as back up systems. The PV panels and co-generation units are described in detail in Appendix A.2, while the other units are described in Appendix A.1. Further details on the formulation can be found in [24]. This optimization is performed to ensure that each building is operated optimally, e.g., all the controllable loads are shifted to decrease the operating cost.

The buildings considered are residential (single- and multi-family houses), mixed (residential and administrative), administrative, commercial, education and hospitals. The buildings are also grouped according to the renovation stage, as existing (built before 2005), new (built after 2005) and renovated (built before 2005 but improved to meet modern standards) [31]. The pool of building meta-models is enriched by including two energy conversion technology configurations, one with and one without $CO_2$ network utilities. Within each scenario, parametric optimisation is implemented on the investment cost (minimum operating cost, minimum investment cost and five intermediary scenarios, see Figure 4) to obtain a systematic approach for generating interesting solutions in cities and explore options for optimal utilities and connections to optimal buildings.

Figure 5 illustrates sample results of the building-level optimisation. More specifically, it depicts the operating-investment cost Pareto frontier and self-sufficiency of residential single-family houses of different renovation stages, with and without $CO_2$ network utilities. The concept of self-sufficiency is further defined in Section 3.5 and is used to evaluate the autonomy of the energy systems studied, but is defined in simple terms as the percentage of electricity consumption supplied by self-production. For all solutions with an increase in investment cost, the operating cost decreases and renewable energy sources penetrate, leading to higher values of self-sufficiency. New and renovated buildings have reduced overall demands, and therefore lower operating cost. While the solutions connected to the $CO_2$ network yield no difference for low investment cost limits, they result in lower operating costs whenever the capital expenditure (capex) limit is high enough for these technologies. However, the piping cost of the $CO_2$ network is not considered at this stage, being included only at the canton/commune level (Switzerland has 26 cantons, each of them being divided in several communes. Cities can be comprised of several communes, e.g., Geneva has four communes).

The building-level solutions are then integrated in the main optimization, where each building is represented by its resource ($CO_2$ liquid and vapor, natural gas, electricity) import and export. Decision variables and constraints are used to permit selection of any number of buildings from any type, age and utility configuration as long as the overall mix is consistent with that of the case study considered. At the city level the optimiser chooses not only the best configuration of buildings, but also additional utilities to create an optimal city. The additional utilities at the upper level include PV panels, central plants which provide $CO_2$ liquid and vapor, and a power-to-gas storage system (Figure 3). The PV panel and $CO_2$ and $CH_4$ storage unit models are described in detail in [25], the co-generation solid oxide fuel cell-gas turbine (SOFC-GT) unit is modeled according to [32] and the co-generation solid oxide electrolysis cell (SOEC) unit according to [33]. A detailed description of the unit models can be found in Appendix A.2 and in [25].

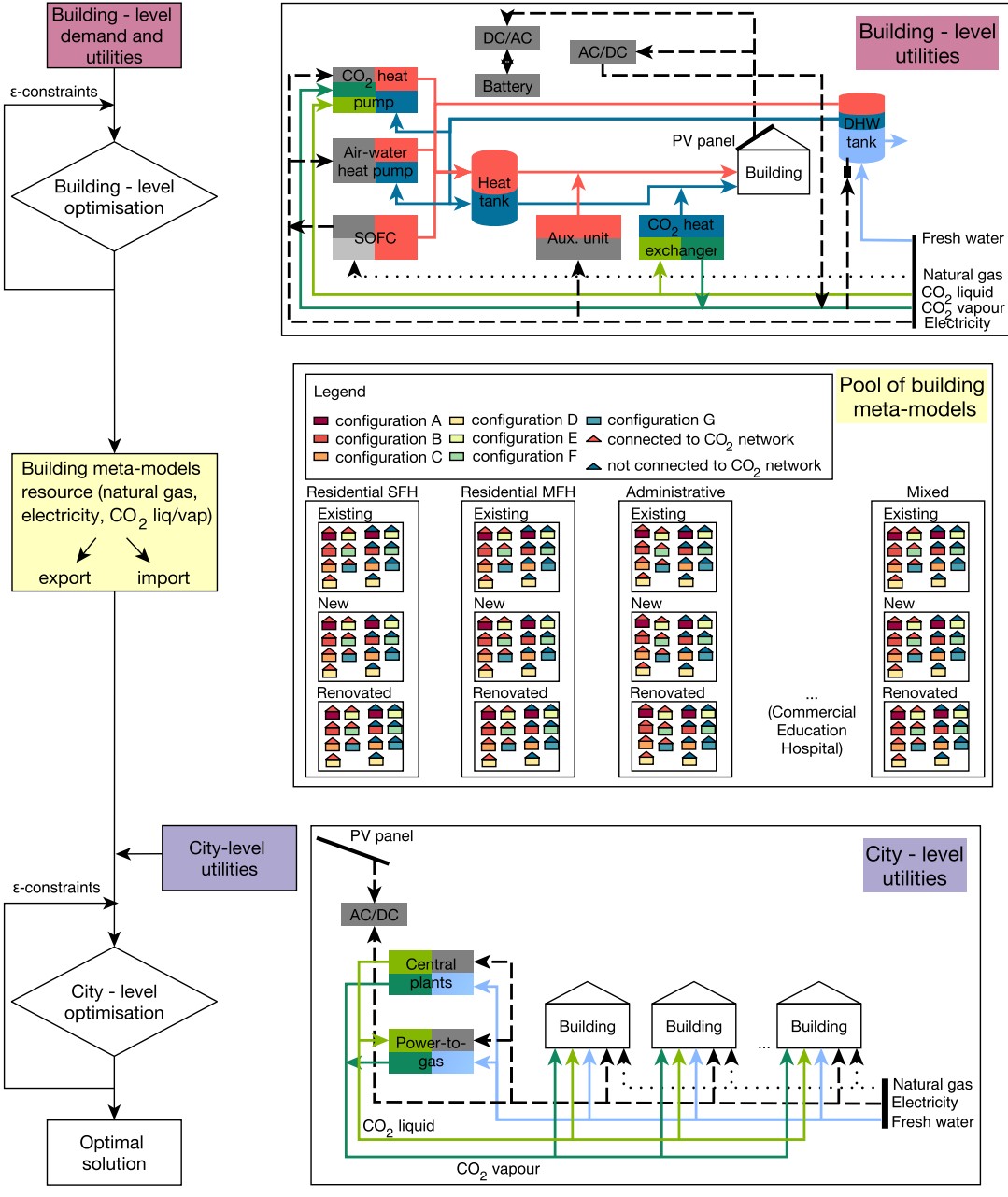

**Figure 3.** Methodology overview.

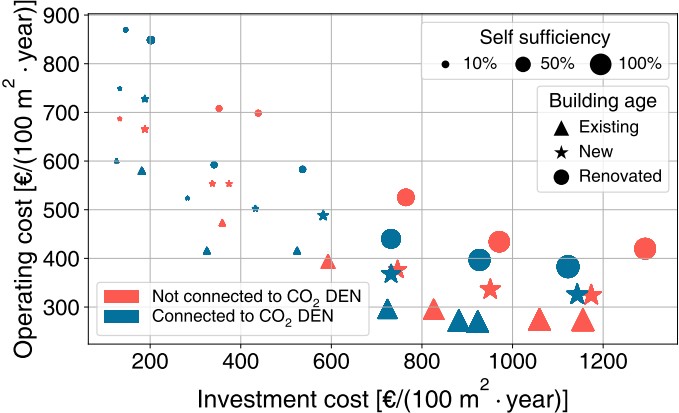

**Figure 4.** Systematic generation method for each building type (i.e., age and renovation).

**Figure 5.** Pareto frontier of residential SFH with different utility configurations and renovation stages.

A mixed integer linear programming (MILP) framework is used to find the optimal utility configurations and to integrate different technologies which satisfy the urban demand.

### 3.2. Mathematical Formulation

The building and city-level optimisations are formulated using mixed integer linear programming [34–37]. This framework was chosen to represent building energy systems, since it can model both the discrete and the continuous behavior of the units. An additional benefit is that this formulation always results in a global optimum and does not require extensive effort for problem initialization.

#### 3.2.1. Definition of Sets

Given that energy demand is time-dependent, the problem is defined using discrete time intervals (e.g., $p \in \mathbf{P} = \{1\}$ (1 year), $t \in \mathbf{TOP}_p = \{1, 2, ..,14\}$ (12 months and two extreme days)). The system to be optimized is represented through several units, belonging to the set $\mathbf{U}$. The units are grouped in two subsets: the set of utility units ($\mathbf{UU}$ = {PV panels, batteries, heat pumps, CHPs, storage tanks, heat exchangers}) and the set of process units ($\mathbf{PU}$ = {building demands: space heating, domestic hot water, air cooling, utilities}). The process units represent the demand and hence have a fixed size, while the utility units represent the energy technologies used to satisfy the demand, with variable

sizes, which are to be optimized. Units supply, demand, or convert resources ($r \in \mathbf{R}$) (electricity and material) and heat (at different temperature intervals $k \in \mathbf{K}$).

### 3.2.2. Objective Function and Constraints

The objective function of the problem is the minimization of the operating cost (Equation (1)), with $\epsilon$-constraints on the investment cost (Equation (2)) [38]. The objective function accounts for both the fixed ($C_u^{op,1}$) and variable ($C_u^{op,2}$) operating costs. The additional terms in the objective function are the binary variables ($y_{u,p,t}$, $y_u$) which decide whether a unit is used or not, the continuous variables ($f_{u,p,t}$, $f_u$) which determine the size of a unit, the operating time parameter ($t_t^{op}$) and the period occurrence ($p_p^{occ}$). $\epsilon$-constraints consider the fixed ($C_u^{inv,1}$) and variable ($C_u^{inv,2}$) investment costs.

$$\min_{y_u, f_u} \sum_{u \in \mathbf{U}} \left( \sum_{p=1}^{\mathbf{P}} \left( \sum_{t=1}^{\mathbf{TOP}} \left( C_u^{op,1} \cdot y_{u,p,t} + C_u^{op,2} \cdot f_{u,p,t} \right) \cdot t_t^{op} \right) \cdot p_p^{occ} \right) \tag{1}$$

$$\sum_{u \in \mathbf{U}} \left( C_u^{inv,1} \cdot y_u + C_u^{inv,2} \cdot f_u \right) \leq \epsilon \qquad \epsilon \in \{IC_{min} : IC_{max}\} \tag{2}$$

The main constraints of the problem include the energy conversion technology sizing and selection. Equations (3)–(6) bound the size of the unit in each time step $t$ and period $p$ to be smaller than the purchase size of the equipment, Equation (7) ensures that the purchase size of the equipment is between the minimum and maximum boundaries set ($f_u^{min}$, $f_u^{max}$), and Equations (8) and (9) fix the size of the process units.

$$y_{u,p} \leq y_u \qquad \forall u \in \mathbf{U}, \ \forall p \in \mathbf{P} \tag{3}$$

$$y_{u,p,t} \leq y_{u,p} \qquad \forall u \in \mathbf{U}, \ \forall p \in \mathbf{P}, \ \forall t \in \mathbf{TOP}_p \tag{4}$$

$$f_{u,p} \leq f_u \qquad \forall u \in \mathbf{U}, \ \forall p \in \mathbf{P} \tag{5}$$

$$f_{u,p,t} \leq f_{u,p} \qquad \forall u \in \mathbf{U}, \ \forall p \in \mathbf{P}, \ \forall t \in \mathbf{TOP}_p \tag{6}$$

$$f_u^{min} \cdot y_{u,p,t} \leq f_{u,p,t} \leq f_u^{max} \cdot y_{u,p,t} \qquad \forall u \in \mathbf{U}, \ \forall p \in \mathbf{P}, \ \forall t \in \mathbf{TOP}_p \tag{7}$$

$$y_{u,p,t} = 1 \qquad \forall u \in \mathbf{PU}, \ \forall p \in \mathbf{P}, \ \forall t \in \mathbf{TOP}_p \tag{8}$$

$$f_u^{min} = f_u^{max} = 1 \qquad \forall u \in \mathbf{PU} \tag{9}$$

The heat cascade equations ensure that heat is transferred from higher temperature intervals to lower temperature intervals and close the energy balance in each temperature interval $k$ (Equation (10a)). This is achieved using the residual heat $\dot{R}_{p,t,k}$, which cascades excess heat from higher temperature intervals ($k$) to lower temperature intervals ($k-1$). The minimum residual heat is zero, when heat cannot be transferred from the corresponding temperature interval to lower ones (Equation (10b)). Similarly, residual heat in the first interval ($\dot{R}_{t,1}$) is zero, as lower temperature intervals do not exist to accept a transfer of heat. Logically, heat cannot be cascaded to the $k^{th}$ interval as it is the highest, so $\dot{R}_{t,k+1}$ is also zero (Equation (10c)). $\dot{Q}_{u,p,t,k}$ represents the reference heat load of a unit $u$ in period $p$, time step $t$ and temperature interval $k$.

$$\sum_{u \in \mathbf{U}} f_{u,p,t} \cdot \dot{Q}_{u,p,t,k} + \dot{R}_{p,t,k+1} - \dot{R}_{p,t,k} = 0 \qquad \forall p \in \mathbf{P}, \ \forall t \in \mathbf{TOP}_p, \ \forall k \in \mathbf{K} \tag{10a}$$

$$\dot{R}_{p,t,k} \geq 0 \qquad \forall p \in \mathbf{P},\ \forall t \in \mathbf{TOP}_p,\ k \in \mathbf{K} \tag{10b}$$

$$\dot{R}_{p,t,1} = 0 \qquad \dot{R}_{p,t,k+1} = 0 \qquad \forall p \in \mathbf{P},\ \forall t \in \mathbf{TOP}_p \tag{10c}$$

For each unit $u$, the supply $\dot{M}^{out}_{r,u,p,t}$ and demand $\dot{M}^{in}_{r,u,p,t}$ of a specific resource $r \in \mathbf{R}$ are computed (Equations (11a) and (11b)) and the balance of each resource is closed for each period $p$ and time step $t$ (Equation (11c)). $\dot{m}^{-}_{r,u,p,t}$ and $\dot{m}^{+}_{r,u,p,t}$ are the reference supply and demand flows of a unit.

$$\dot{M}^{-}_{r,u,p,t} = \dot{m}^{-}_{r,u,p,t} \cdot f_{u,p,t} \qquad \forall r \in \mathbf{R},\ \forall u \in \mathbf{U},\ \forall p \in \mathbf{P},\ \forall t \in \mathbf{TOP}_p \tag{11a}$$

$$\dot{M}^{+}_{r,u,p,t} = \dot{m}^{+}_{r,u,p,t} \cdot f_{u,p,t} \qquad \forall r \in \mathbf{R},\ \forall u \in \mathbf{U},\ \forall p \in \mathbf{P},\ \forall t \in \mathbf{TOP}_p \tag{11b}$$

$$\sum_{u \in \mathbf{U}} \dot{M}^{-}_{r,u,p,t} = \sum_{u \in \mathbf{U}} \dot{M}^{+}_{r,u,p,t} \qquad \forall r \in \mathbf{R},\ \forall p \in \mathbf{P},\ \forall t \in \mathbf{TOP}_p \tag{11c}$$

### 3.2.3. Constraint Linking Individual Building and Urban Scale

Specific constraints at the building scale are presented in detail in [24]. Additional variables and sets are introduced at the urban scale, which aid the formulation of the constraints, such as building types ($bt \in \mathbf{BT}$ = {residentialSFH, residentialMFH, administrative, education, commercial, hospital, mixed}), building units of type $bt$ ($bu \in \mathbf{BUT}_{bt}$), renovation stages ($rs \in \mathbf{RS}$ = {existing, new, renovated}), building units of renovation stage $rs$ ($bu \in \mathbf{BUR}_{rs}$), building units connected to the $CO_2$ network ( $bu \in \mathbf{BUC}$) and the set of cities/communes ($c \in \mathbf{C}$). The extra constraints include fixing the number of buildings of a given type to the one of the case studies considered ($N_{bt}$):

$$\sum_{p=1}^{\mathbf{P}} \sum_{t=1}^{\mathbf{TOP}} \sum_{bu \in \mathbf{BUT}_{bt}} f_{bu,p,t} = N_{bt} \qquad \forall bt \in \mathbf{BT} \tag{12}$$

And making the number of buildings at each renovation stage equal to that of the urban system studied ($N_{rs}$):

$$\sum_{p=1}^{\mathbf{P}} \sum_{t=1}^{\mathbf{TOP}} \sum_{bu \in \mathbf{BUR}_{rs}} f_{bu,p,t} = N_{rs} \qquad \forall rs \in \mathbf{RS} \tag{13}$$

The investment cost for the $CO_2$ network in each city/commune is computed according to [31] (for details see Appendix A.3). The commune has the choice of investing in the $CO_2$ network or not ($y_{u_{CO_2,c}}$), which translates in optimisation terms as a big $M$ constraint:

$$y_{u_{CO_2,c}} \geq f_{bu,p,t}/M \qquad \forall bu \in \mathbf{BUC}_c,\ \forall c \in \mathbf{C},\ \forall p \in \mathbf{P},\ \forall t \in \mathbf{TOP}_p \tag{14}$$

I.e., if the commune activates a building with $CO_2$ network utilities, it must invest in piping. The size/length of the network piping is fixed for all periods and times.

### 3.2.4. Long-Term Energy Storage Model with Typical Day Resolution

To model the long-term storage units with typical day resolution, a series of new sets must be introduced (or re-defined). The equations here are based on using eight periods or typical days:

- **P**: periods, or typical days of the year, e.g., {1, 2, 3, 4, 5, 6, 7, 8};
- **TOP**$_p$, $\forall p \in \mathbf{P}$: time steps in each period $p$, e.g., $\underbrace{\{\{1, 2, ..., 24\}, \{25, 26, ..., 48\}, ..., \{169, 170, ..., 192\}\}}_{\text{8 typ. days}}$;

- **RD**: real days of the year, e.g., {1, 2, ..., 365};

- **PORD**$_{rd}$, $\forall rd \in \mathbf{RD}$: typical day corresponding to each real day of the year, e.g., $\underbrace{\{2, 2, 4, ...6\}}_{365 \text{ days}}$;

- **TORD**$_{rd} = t$, $\forall t \in \mathbf{TOP}_{pr}$, $\forall pr \in \mathbf{PORD}_{rd}$, $\forall rd \in \mathbf{RD}$: time steps in each real day of the year, e.g., $\underbrace{\{\underbrace{\{25, 26, ..., 48\}}_{\text{time steps in typ. day 2}}, \underbrace{\{25, 26, ..., 48\}}_{\text{time steps in typ. day 2}}, \underbrace{\{73, 74, ..., 96\}}_{\text{time steps in typ. day 4}}, ..., \underbrace{\{121, 122, ..., 144\}}_{\text{time steps in typ. day 6}}\}}_{365 \text{ days}}$;

- **TOPNC**$_p = \{1, 2, ..., \text{card}(\mathbf{TOP}_p)\}$, $\forall p \in \mathbf{P}$: non cumulative time steps in each typical day $p$, e.g.,

  $\underbrace{\{\{1, 2, ..., 24\}, \{1, 2, ..., 24\}, ..., \{1, 2, ..., 24\}\}}_{8 \text{ typ. days}}$;

- **RTORD**$_{rd,pr,t} = \sum_{i=1}^{rd-1} \text{card}(\mathbf{TORD}_i) + t - \sum_{j=1}^{pr-1} \text{card}(\mathbf{TOPNC}_j)$ $\forall t \in \mathbf{TOP}_{pr}$, $\forall pr \in \mathbf{PORD}_{rd}$, $\forall rd \in \mathbf{RD}$: real time of each real day of the year, e.g.,

  $\underbrace{\{\underbrace{\{1, 2, ..., 24\}}_{\text{time steps in real day 1}}, \underbrace{\{25, 26, ..., 48\}}_{\text{time steps in real day 2}}, \underbrace{\{49, 50, ..., 72\}}_{\text{time steps in real day 3}}, ..., \underbrace{\{8737, 8738, ..., 8760\}}_{\text{time steps in real day 365}}\}}_{365 \text{ days}}$

- **RT** $= 1, ..., \sum_{rd \in \mathbf{RD}} \text{card}(\mathbf{TORD}_{rd})$, (ordered set): real times of the year, e.g., {1, 2, ..., 8760}.

  Given these sets above, the long-term storage units ($u \in \mathbf{SU}$) are represented by the constraint:

$$SL_{rt} = \begin{cases} \text{if } rt = \text{first}(\mathbf{RT}): & \sigma \cdot SL_{\text{last}(\mathbf{RT})} + \eta_{ch} \cdot M^+_{r,u,pr,t} - \frac{1}{\eta_{dch}} \cdot M^-_{r,u,pr,t} \\ \text{else}: & \sigma \cdot SL_{rt-1} + \eta_{ch} \cdot M^+_{r,u,pr,t} - \frac{1}{\eta_{dch}} \cdot M^-_{r,u,pr,t} \end{cases}$$

$$\forall r \in \mathbf{R}, \forall u \in \mathbf{SU}, \forall rd \in \mathbf{RD}, \forall pr \in \mathbf{PORD}_{pr}, \forall t \in \mathbf{TOP}_{pr}, \forall rt \in \mathbf{RTORD}_{rd,pr,top} \quad (15)$$

with $SL_{rt}$ as the storage level of the unit at each real time step of the year $rt \in \mathbf{RT}$, $\sigma = 0.9992$ [39] the self-discharge rate of the unit, and $\eta_{ch} = \eta_{dch} = 0.9$ [39] as the charging and discharging efficiencies of the unit. A summary of all the sets used in the problem formulation is given in Table 1.

**Table 1.** Sets used in the mathematical formulation.

| Set Symbol | Name | Index | Increment | Cyclicity |
|---|---|---|---|---|
| **P** | periods | - | day | no |
| **TOP** | times of period | $p$ | hour | no |
| **U** | units | - | - | no |
| **UU** | utility units | - | - | no |
| **PU** | process units | - | - | no |
| **SU** | storage units | - | - | no |
| **R** | resources | - | - | no |
| **K** | temperature intervals | - | - | no |
| **BT** | building types | - | - | no |
| **BUT** | building units of type | $bt$ | - | no |
| **RS** | renovation stages | - | - | no |
| **BUR** | building units of renovation | $rs$ | - | no |
| **C** | communes | - | - | no |
| **BUC** | building units connected to $CO_2$ DEN | $c$ | - | no |
| **RD** | real days | - | day | no |
| **PORD** | periods of real day | $rd$ | day | no |
| **TORD** | times of real day | $rd$ | hour | no |
| **TOPNC** | times of period non cummulative | $pr$ | hour | no |
| **RTORD** | real times of real day | $rd, pr, t$ | hour | no |
| **RT** | real times | - | hour | yes |

### 3.3. Case Study

The case studies considered are Geneva city center (four communes: Genève-Cité, Genève-Plainpalais, Genève-Eaux-Vives and Genève-Petit-Saconnex) and the canton of Geneva (all 48 communes, Figure 6).

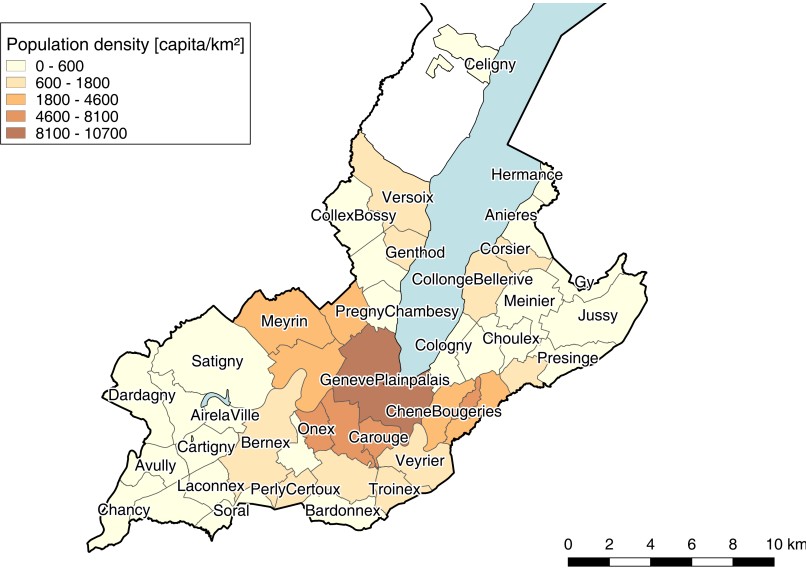

**Figure 6.** Population density of communes in the canton of Geneva.

The building types are distinguished according to the RegBL database [40], as listed in Table 2. The corresponding parameter names in the RegBL report are listed in Table A18 in the Appendix A.4.

**Table 2.** Building types present in the model of the canton of Geneva.

| Building Type | Building Category | Building Class |
|---|---|---|
| Residential SFH | 1021, 1025, 1230 | 1110 |
| Residential MFH | 1025 | 1121 |
| | 1040 | 1130 |
| Administrative | 1040, 1060 | 1220 |
| Commercial | 1040, 1060 | 1230 |
| Education | 1040, 1060 | 1263 |
| Hospital | 1040, 1060 | 1264 |
| Mixed | 1030 | 1121, 1122 |

The energy reference area (ERA) of the buildings ($A_b^{ERA}$) is computed according to the same database, using the footprint area of the building ($A_b$), the number of floors ($N_b^{floors}$) and a factor of 0.9, an assumption used to account for the inner walls (Equation (16)).

$$A_b^{ERA} \; [m^2] = A_b \; [m^2] \cdot N_b^{floors} \cdot 0.9 \tag{16}$$

The photo-voltaic rooftop potential is calculated using the rooftop area of the building ($A_b^{roof}$), the average solar irradiation on each roof ($I_b$), a nominal global horizontal irradiation of 1244.334 W/(m²·K) and a factor of 0.75 to account for the part of the roof which cannot be covered with PV panels (e.g., close to the periphery) (Equation (17) [41]).

$$A_{PV,b} \; [m^2] = A_b^{roof} \; [m^2] \cdot \frac{I_b \; [W/(m^2 \cdot K)]}{1244.334 \; [W/(m^2 \cdot K)]} \cdot 0.75 \tag{17}$$

The number of buildings of each category and renovation stage are considered according to [31] and Figure 7 displays a sample distribution, that of Geneva city center.

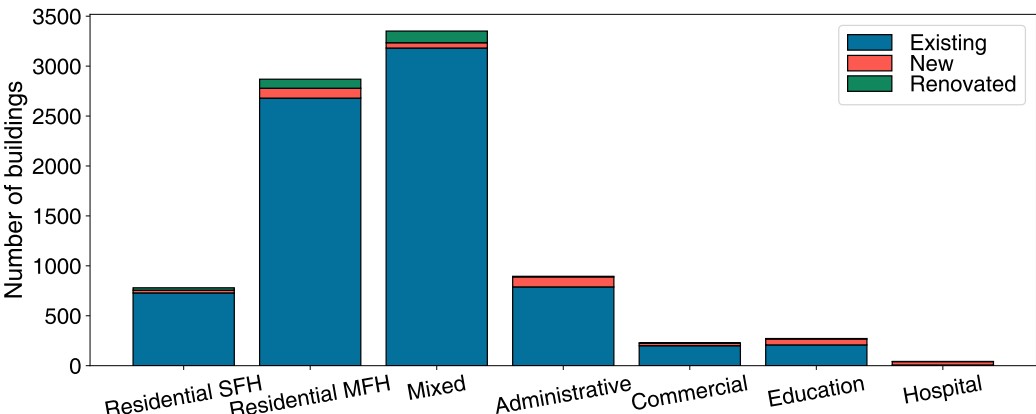

**Figure 7.** Refurbishment level building distribution in Geneva city center.

The demands evaluated are space heating, domestic hot water, air cooling and electricity. The hourly demand profiles are built based on standards and existing heat signature models. The electricity and domestic hot water demand profiles are considered according to the standards of the Swiss society of engineers and architects SIA [42] with a typical day profile  repeated throughout the year, while the heating and cooling demands are modeled based on a heating signature profile [31].  These profiles have been calibrated based on statistical data from the energy department of the canton of Geneva [31].  Figure 8, Table 3 and Table A25 display the hourly demand profile of administrative buildings (existing, new and renovated) and their specific yearly demand. The domestic hot water and electricity demand is constant; therefore it is excluded from the hourly variation plots. The corresponding plot/table for all other building categories can be found in Appendix A.5.

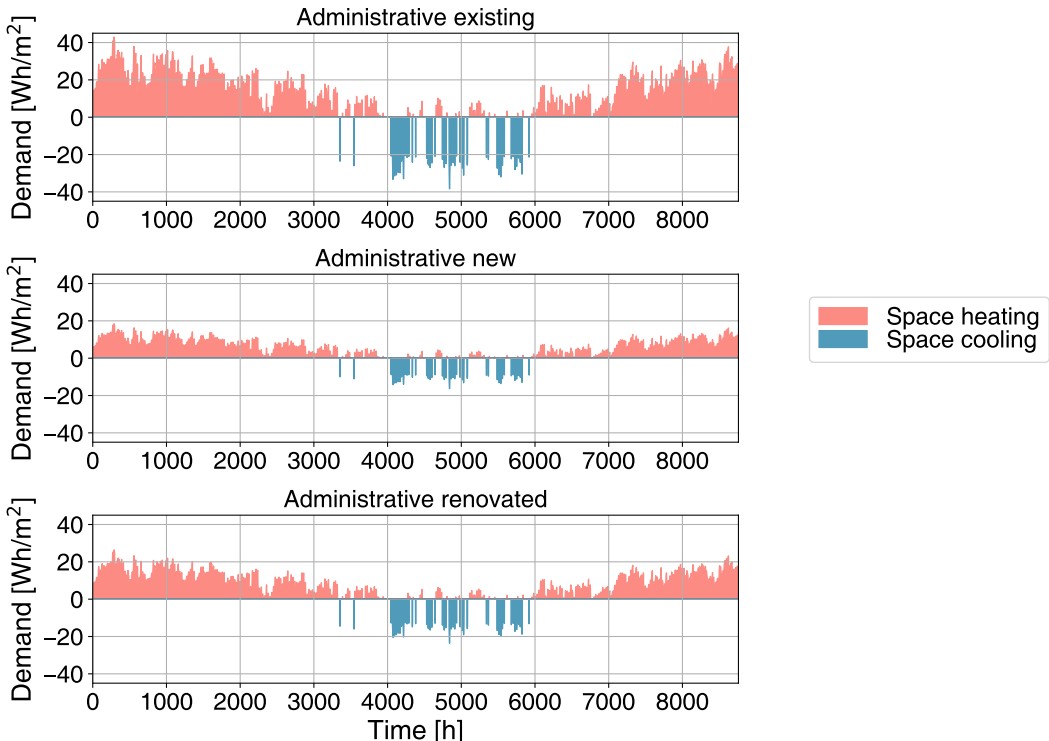

**Figure 8.** Hourly specific energy service demand of administrative buildings.

**Table 3.** Yearly specific energy service demand of administrative buildings.

| Building Renovation Stage | Space Heating [kWh/m$^2$] | Air Cooling [kWh/m$^2$] | Dom. Hot Water [kWh/m$^2$] | Electricity [kWh/m$^2$] |
|---|---|---|---|---|
| Existing | 84.2 | 8.9 | 2.9 | 43.2 |
| New | 36.4 | 3.8 | 2.9 | 43.2 |
| Renovated | 51.7 | 5.5 | 2.9 | 43.2 |

### 3.4. Time Resolutions: Typical Days Algorithm

Two time resolutions are used to solve the optimization problem, namely the state-of-the-art monthly averages with two extreme periods, and hourly resolution. Since the computational time for solving the problem increases drastically with the problem size (Figure 9), a k-medoids-based data clustering algorithm is used to reduce the complexity of the problem studied (Figure 10) for hourly resolution. This approach selects the cluster centers based on the smallest sum of distances within each cluster, while the cluster size is selected based on a series of performance indicators [43,44].

Two input parameters are considered for the clustering process, namely the ambient temperature ($T_{ext}$) and the global solar irradiation (GI), since all resources and demands can either be computed using these two parameters, or are assumed constant. Other data such as consumption profiles and their corresponding temperatures of demand are defined based on the computed cluster centers. The k-medoids algorithm is applied between 2 and 25 typical days. A maximum of 12% error in the load duration curve (ELDC) is set and consequently the number of typical days should be greater than five (Figure 9). To select the optimal number of typical days, the Davies-Bouldin (DB) index is used. The DB index is a measure of clustering scheme performance [45]. It accounts for the separation between the clusters—which should be as large as possible—and the within-cluster scatter, which should be as low as possible. The index is defined as the ratio between the cluster separation and the within-cluster distance, where lower values express better cluster separation and the 'tightness' inside the clusters. As observed in Figure 9, the DB index has the lowest value for 8 typical days for the dataset studied here. Therefore, this value is used for further analysis.

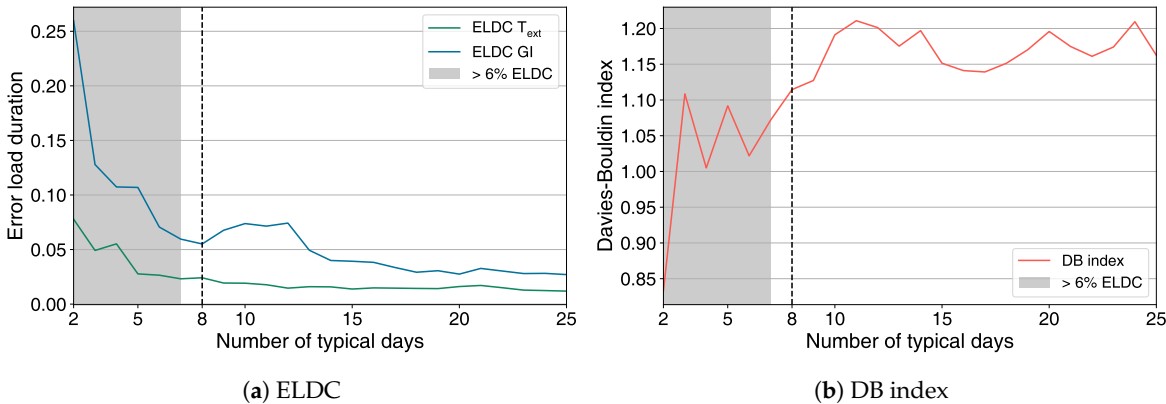

(**a**) ELDC  (**b**) DB index

**Figure 9.** Performance indicator evolution using the k-medoids algorithm for selecting the number of typical days.

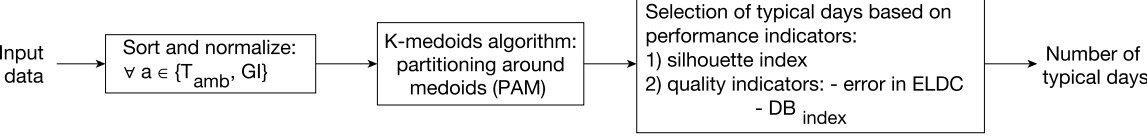

**Figure 10.** Typical days algorithm.

Figure 11 depicts the real profile of the two attributes chosen to cluster the data in grey the computed load duration curve in black. One can see that the load duration curve of both attributes is followed well with the number of typical days chosen.

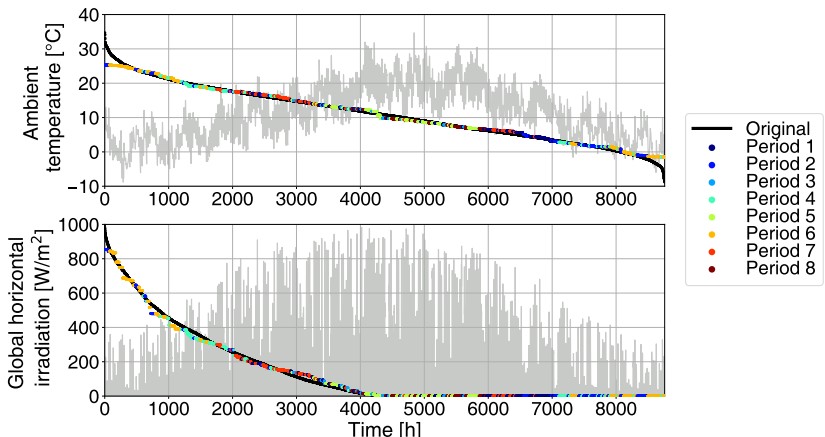

**Figure 11.** Representation of the typical days algorithm.

To clarify contributions of the different time resolutions in the problem formulation, the objective function (Equation (1)) can be assessed in greater detail. For monthly resolution, $p = 1$ (1 year), $t = \{1, 2, ..., 14\}$ represents 12 months and 2 extreme periods, $p_p^{occ} = 1$ represent the occurrence of the year, and $t_t^{op} = \{744, 672, 744, ..., 744, 0, 0\}$ are the number of operating hours in each time step $t$. With hourly resolution, $p = \{1, 2, ..., 10\}$ are the eight typical days and the two extreme hours, $t = \{24, 24, ..., 24, 1, 1\}$ are the number of hours in each time step $t$, $p_p^{occ} = \{54, 46, 17, 49, 52, 68, 49, 30, 1, 1\}$ represents the number of times each operating period appears during the year, and $t_t^{op} = \{1, 1, ..., 1, 0, 0\}$ is the operating time of each time step. For both time resolutions, the operating time of the extreme periods is zero, since they are used only for unit sizing.

## 3.5. Measure of Energy Autonomy

In this work, a urban community is considered energy autonomous when the electricity import from the grid ($E_i$) is zero, or likewise, when the self-sufficiency (SF) factor (Equation (18) [46]) is equal to unity. A solution is considered to be net zero-energy when the power grid export ($E_e$) and import ($E_i$) are equal, which is equivalent to when the self-sufficiency factor (Equation (18)) equals the self-consumption (SC) factor (Equation (21), Figure 12), where $E_g$ represents the electricity generation (e.g., by PV panels, co-generation units).

$$SF = \frac{E_g - E_e}{E_g - E_e + E_i} \tag{18}$$

where the numerator represents the demand:

$$E_g - E_e + E_i = \sum_{p=1}^{P} \sum_{t=1}^{TOP} \left( \dot{M}^+_{el,elheater,p,t} + \dot{M}^+_{el,battery,p,t} + \dot{M}^+_{el,HPs+Ref,p,t} + \right.$$

$$\left. \dot{M}^+_{el,House,p,t} + \dot{M}^+_{el,CPwinter,p,t} + \dot{M}^+_{el,SOEC,p,t} \right) \tag{19}$$

and the electricity generation is given by:

$$E_g = \sum_{p=1}^{P} \sum_{t=1}^{TOP} \left( \dot{M}^-_{el,PV,p,t} + \dot{M}^-_{el,SOFC,p,t} + \dot{M}^-_{el,Battery,p,t} \right) \tag{20}$$

$$SC = \frac{E_g - E_e}{E_g} \tag{21}$$

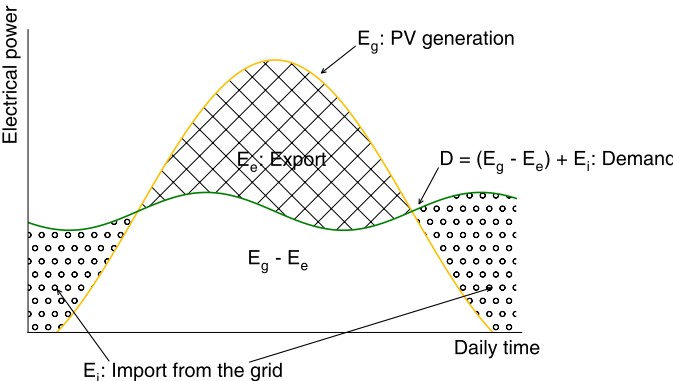

**Figure 12.** Self-sufficiency and self-consumption visual depiction.

## 4. Results and Discussion

### 4.1. Monthly vs. Typical Day Time Resolution

First, the two time resolutions considered (average monthly and typical day hourly) are analyzed for the case study of Geneva city center. Figure 13 depicts the operating-investment cost Pareto front for the two time resolutions, the size of the dots represents the self-sufficiency of the system and the solutions connected to the $CO_2$ network are highlighted in gray. By comparing the two time resolutions, it is observed that for the same investment cost limits, solutions using monthly resolution yield up to 31% lower operating cost, and 18% higher self-sufficiency (for the 8th investment cost limit). This occurs due to the fact that peak shaving is an implicit outcome of data aggregation for the monthly resolution, while peaks must be accounted for explicitly with the hourly resolution and adjustments must be made to buy electricity even when previous electricity sales may have occurred. This results in higher operating cost and lower self-sufficiency by considering scenarios with hourly profiles. This also stresses the importance of considering analysis with enough temporal detail to understand the real system requirements, since grid balancing must be completed on short time scales and thus analysis using average data may lead to problematic scenarios.

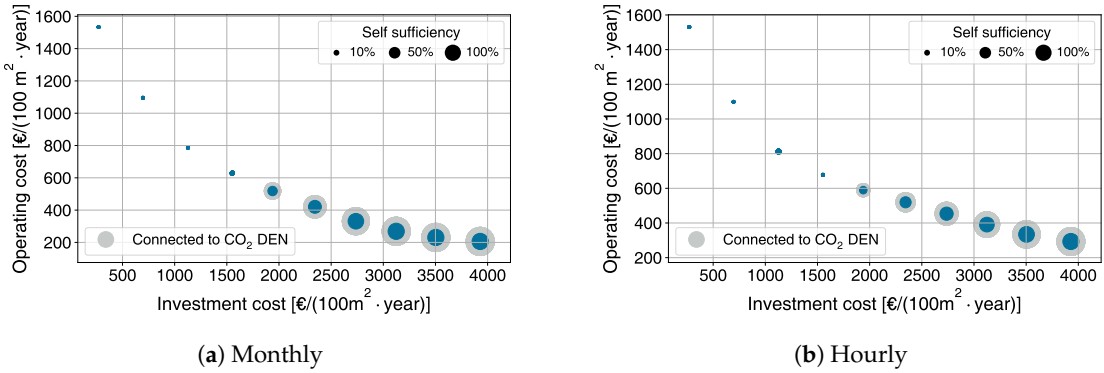

(**a**) Monthly      (**b**) Hourly

**Figure 13.** Pareto Geneva city center different time resolutions.

The cost breakdown of the two time resolutions is shown in Figure 14. The first figures on the left show the breakdown of total cost, the biggest contribution being the capex since the system starts optimal solutions require increasing investment to reduce the operating cost and increase the self-sufficiency. A high level of investment is required to supply the peak demand; however, investing

approximately 60% of the maximum value yields solutions with self-sufficiency in excess of 60%. The second and third set of figures, the breakdown of investment cost at the building and city levels, show that both time resolutions highlight the same main contributors: heat pumps, SOFCs and PV panels at the building level and PV panels, power-to-gas and the $CO_2$ network pipes at the city level.

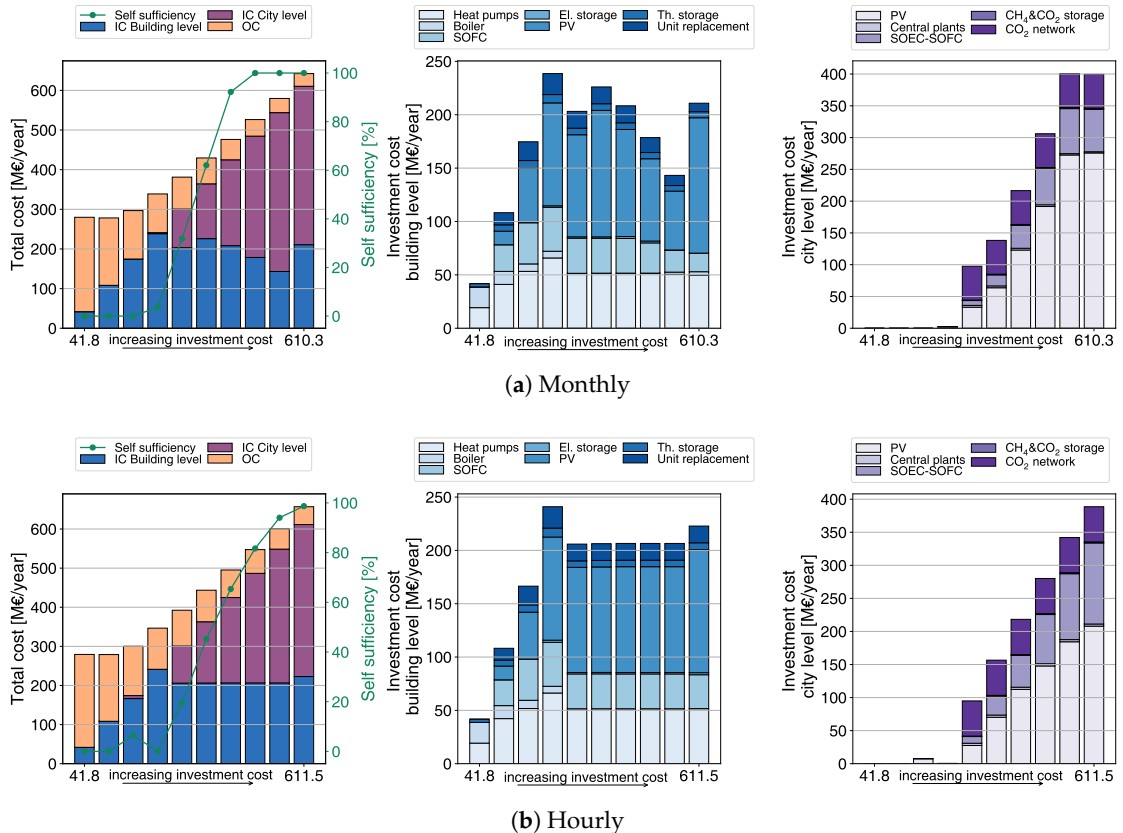

**Figure 14.** Geneva city center cost breakdown for different time resolutions.

As shown in these results, both time resolutions show the same trends and main cost contributors. Therefore, despite the increased accuracy provided by hourly resolution and in the interest of reducing data processing and computation time, the remaining results, at canton level (48 communes), are obtained using a monthly resolution.

### 4.2. $CO_2$ DEN Penetration with Population Density

An $\epsilon$-constraint investment cost optimisation for all 48 communes in the canton of Geneva is performed to study the $CO_2$ DEN penetration depending on the population density. The investment cost of the $CO_2$ network was considered to be explained in Equation (14). Figure 15 depicts the lowest investment cost, 40% of maximum IC, 90% of maximum IC and lowest operating cost scenarios. This figure shows that the scenario with the lowest investment cost does not prompt any of the communes to invest in the $CO_2$ network. With increasing investment cost limits, the communes which are most densely populated, with an investment cost per energy reference area lower than 21.5 k€/100 m$^2_{ERA}$, start connecting to the $CO_2$ DEN. Finally, for the minimum operating cost scenario, all communes connect to the low temperature DEN.

The results are also represented using parallel coordinates. Figure 16 shows that higher investment cost limits logically correlate with reduced operating cost and $CO_2$ emissions in the canton. Moreover, higher overall investment cost solutions lead to the largest number of communes connected to the $CO_2$ DEN and the highest self-sufficiency of the canton. Regarding the investment cost at the building level, a mix of high and low investment cost buildings are selected for optimal operating cost, with a

moderate investment in PV panels and heat pumps. The solution with the lowest operating cost is selected to explore detailed results, as highlighted in Figure 16. Compared with the current situation (i.e., lowest investment cost solution: mostly boilers supplying heating, no PV market penetration), the best scenario (from an economic standpoint) leads to approximately 90% savings in $CO_2$ emissions and operating cost, with a payback time of 17.5 years.

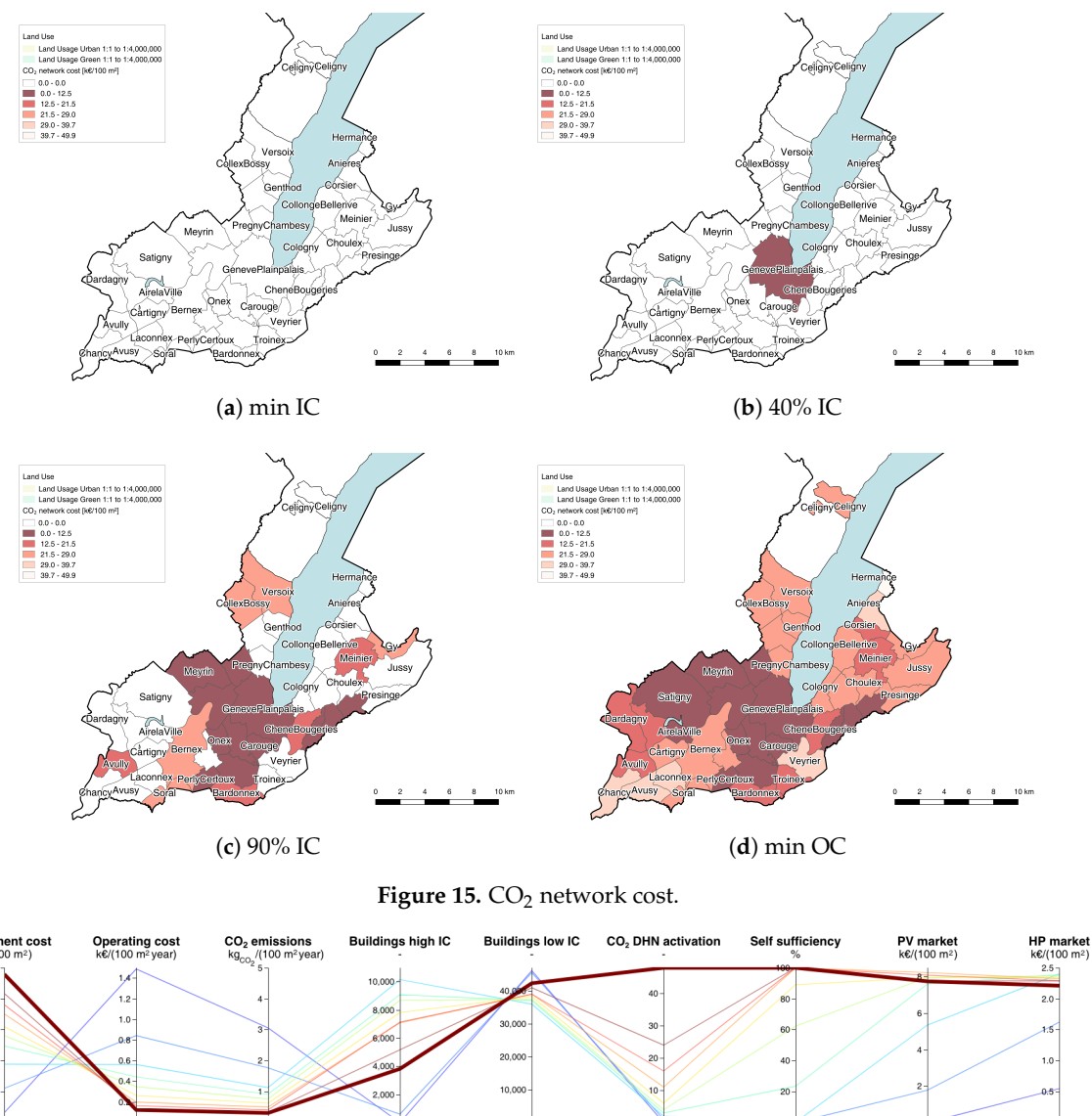

**Figure 15.** $CO_2$ network cost.

**Figure 16.** Parallel coordinate representation of the canton solutions.

*4.3. Detailed Results of Solution with Lowest Operating Cost and Emissions*

Figure 17 depicts the details of the solution highlighted above, for each of the 48 communes, sorted by population density. Most of the communes have low population and building densities, and correspondingly low energy flows (i.e., electricity and natural gas import/export). Generally, high population densities are associated with lower district network cost per energy reference area and with high $CO_2$ emissions. However, the environmental impact has a higher correlation with the overall population, i.e., with the total electricity and natural gas import.

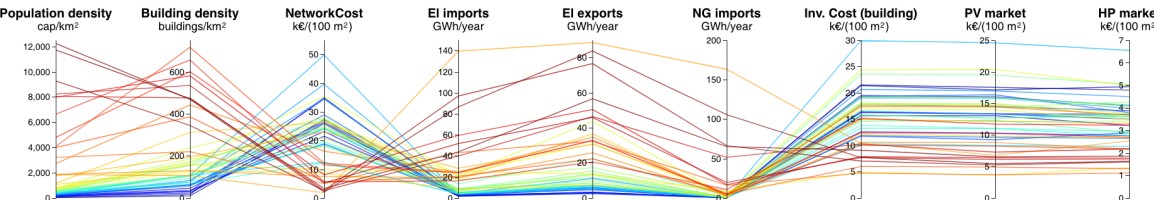

**Figure 17.** Parallel coordinate representation of the communes for the lowest operating cost solution.

Figure 18 is used to detail the energy flows in the communes, by displaying the detailed contributors of electricity and natural gas import/export at the building level for each commune and at the canton level. The results show that:

- the main electricity consumers at the building level are heat pump and refrigeration units ($\approx$35%) and electrical appliances ($\approx$65%);
- the main electricity producers are PV panels, accounting for 91% of the production and SOFC co-generation units supplying the balancing 9%;
- the main natural gas consumers are boilers (47%) and SOFC co-generation units (53%).

At the cantonal level:

- electricity is consumed by the SOEC unit (35%), by the central plants to produce $CO_2$ (9%) vapour, and by net electricity importing buildings (56%);
- electricity is produced by PV panels (77%) and the SOFC co-generation unit (23%);
- natural gas is required for the SOFC unit (18%) and for net natural gas importing buildings (82%);
- natural gas is produced by the methanation unit in the power-to-gas system (18%) and purchased from the grid (82%).

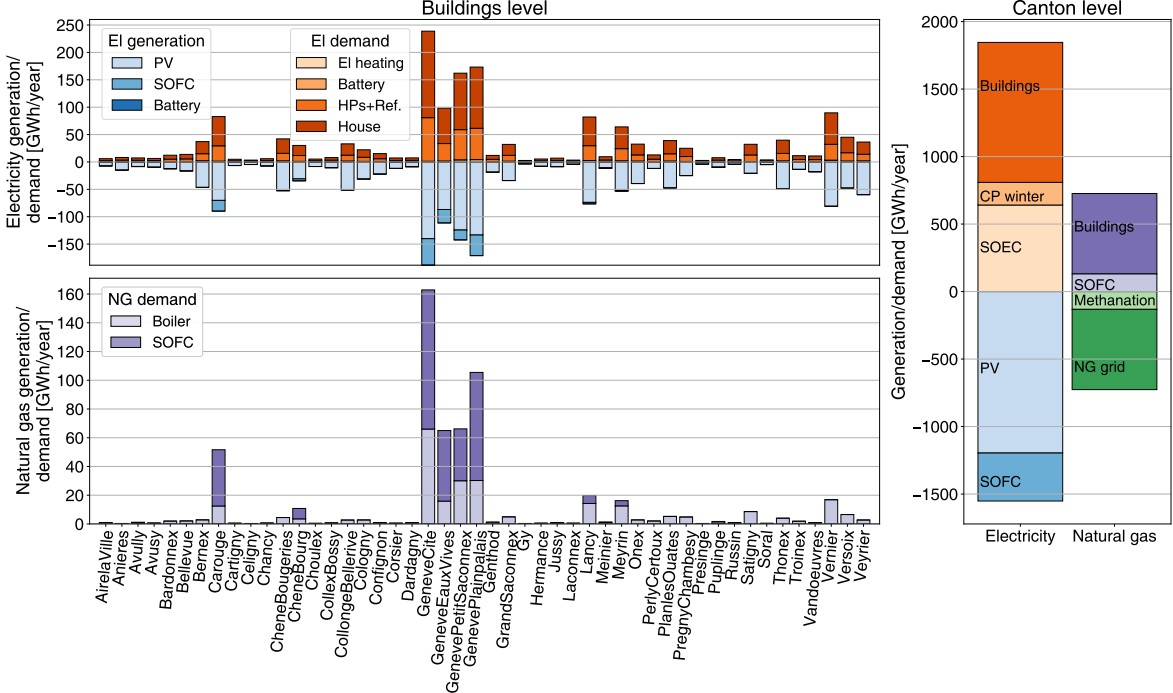

**Figure 18.** Breakdown of electricity/natural gas import/export by commune and at building/canton level.

Figure 19 shows the cost breakdown at the building level, for each commune and at the canton level. Similar to the results shown for Geneva city center, building invesments are principally concentrated in heat pumps and refrigeration units ($\approx$20%), SOFCs ($\approx$3%) and PV panels ($\approx$71%),

while the investment cost at the canton level is dominated by the $CO_2$ DEN piping (28%) followed by PV panels (19%) and the the power-to-gas system (9%).

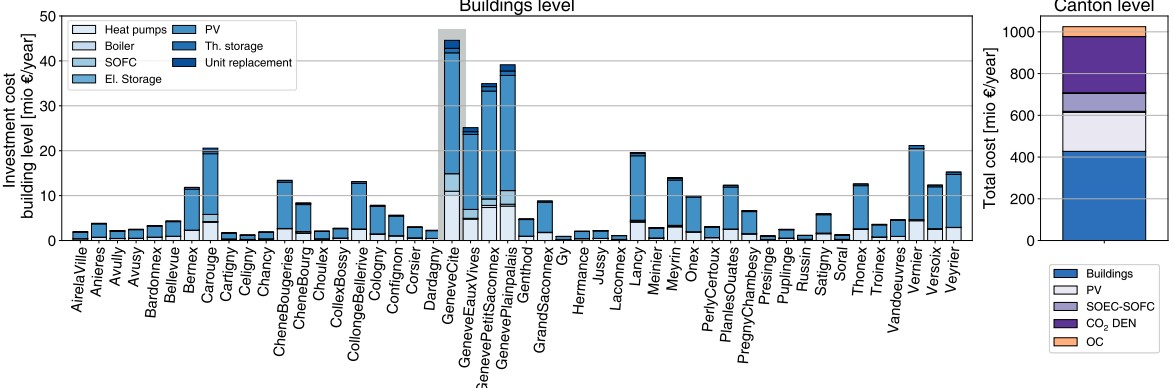

**Figure 19.** Breakdown of investment and total cost at buildings/canton level.

Also in this case one of the solutions, the one of the commune Génève Cité (highlighted solution in Figures 17 and 19), is selected for additional exploration.

The monthly energy import/export profiles of Génève Cité are shown in Figure 20. As observed in this figure, the electricity consumption of heat pumps is high in winter, when heating is required, and lower in summer, while the electricity demand for electrical appliances is assumed constant over the year. Electricity production from PV panels is higher in summer, corresponding to higher global horizontal irradiation, and the electricity production of SOFC co-generation units is higher in winter, since they provide the electricity requirement of heat pumps and co-generate heat for space heating and domestic hot water demand. Consequently, the natural gas consumption of the SOFC and boiler units are higher in winter, both related to supply of heating services.

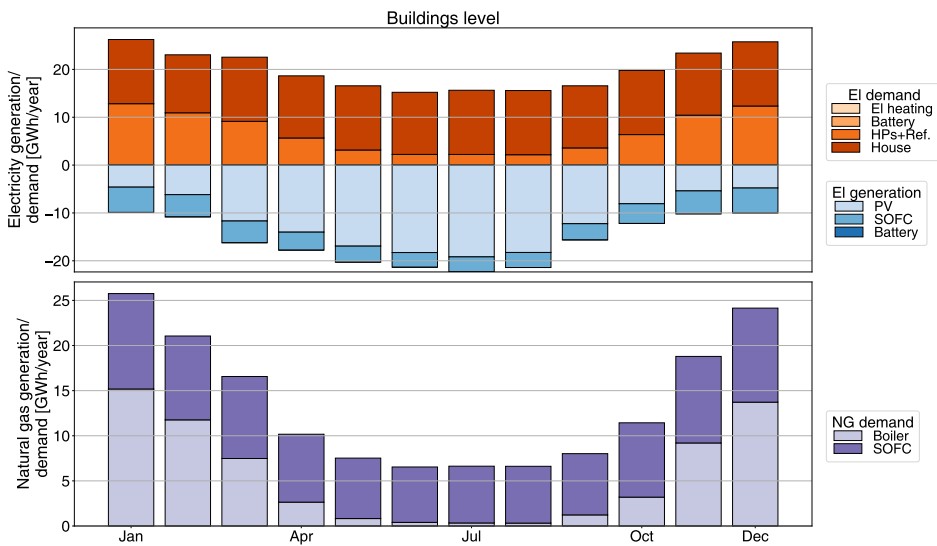

**Figure 20.** Breakdown of electricity/natural gas import/export by month, for Génève Cité.

## 5. Conclusions

This paper aims at providing a method to systematically integrate multi-energy networks and low carbon resources in cities. The method proposes a double optimisation approach with meta-models to link analysis and optimisation at both building and urban scales. The first optimisation creates a pool of optimal building solutions of different types (residential single- and multi-family houses (SFH, MFH), administrative, commercial, education, hospital and mixed), renovation stages (existing, new and

renovated), energy system configurations (existing $H_2O$-based networks or potential low-temperature $CO_2$-based networks) and for different investment cost limits (i.e., budgets). The pool of optimal building meta-models is fed to the optimiser on top, which selects the best mix to optimise energy systems at city/canton level.

Geneva city center is used as a case study to analyse the impact of different temporal resolutions, namely monthly averages and hourly typical days. The results show that implicit peak shaving occurs in the monthly resolution by averaging demands, resulting in lower operating cost and higher self-sufficiency solutions, compared to the hourly resolution. However, the investment cost breakdown proves that the main contributors do not change, irrespective of the time resolution. Therefore, and in view of decreasing data processing and computation time, a monthly resolution is used for the results at canton level.

The second case study, the whole canton of Geneva (48 communes), is analysed to assess $CO_2$ DEN penetration with population density. The results highlight that scenarios with moderate investment limits, only communes with high population density, i.e., a network cost below 21.5 k€/100 $m^2_{ERA}$ connect to the refrigerant-based network, while for the minimum operating cost scenario, all communes are connected to the $CO_2$ DEN. Parallel coordinates are employed to better visualise key performance indicators for the scenarios at the cantonal and communal levels. The energy and cost breakdown results for each commune show that electricity is mostly consumed in heat pumps, refrigeration units and for electrical appliances while being produced by PV panels and SOFC co-generation units, while natural gas is consumed for boilers and SOFCs. Consequently, at the building level, the investment cost is dominated by heat pumps ($\approx$20%), SOFCs ($\approx$3%) and PV panels ($\approx$71%). At the canton level, the electricity importers are the buildings, SOEC unit and central plant, and the electricity exporters are PV panels and the SOFC unit, while the natural gas importers are the building and SOFC unit and the exporters are the methanator and natural gas grid. Consequently, the investment cost at the cantonal level is dominated by PV panels (19%), the power-to-gas system (9%) and $CO_2$ DEN piping (28%).

This work successfully develops an integrated framework, which embeds optimally operating buildings in districts. The framework was validated using the canton of Geneva; however, it is not case specific and can therefore be applied to different urban systems/conditions. This work allows engineers to assess the cost of reaching the Paris agreement targets and reduce the operating cost by approximately 90% in the residential sector, while using low-temperature $CO_2$ district energy networks. The model can also be used to study the integration of other types of large energy systems, e.g., by municipal bodies for future planning of urban energy supply with long planning horizons. Future work includes improving the pool of building meta-models, to cover a wider range of building types and a finer resolution on the building renovation stage and on budget scenarios. A typical day/full hourly resolution is suggested for future work to obtain more precise results and avoid inaccuracies stemming from implicit peak shaving. Further applications of the method in other geographical contexts would create a broader understanding of $CO_2$ DEN penetration and could potentially be extended to a European or global scale to assess feasibility as a multi-energy, fully renewable solution, coupled to long-term energy storage.

**Author Contributions:** Conceptualization, R.S., P.S., I.K., L.G. and F.M.; Formal analysis, R.S., P.S. and I.K.; Investigation, R.S., P.S., I.K., L.G. and F.M.; Methodology, R.S. and P.S.; Project administration, F.M.; Resources, F.M.; Software, R.S. and P.S.; Supervision, I.K. and F.M.; Validation, R.S., P.S. and I.K.; Visualization, R.S., P.S., I.K., L.G. and F.M.; Writing—original draft, R.S.; Writing—review & editing, R.S., P.S., I.K. and L.G.

**Funding:** This project is carried out with the financial support of the Swiss Innovation Agency (Innosuisse-SCCER program).

**Acknowledgments:** This project is carried out with the financial support of the Swiss Innovation Agency (Innosuisse - SCCER program) and of the Swiss Federal Office of Energy in the context of the ERA-NET Co-fund Smart Cities and Communities (ENSCC) project IntegrCiTy, with financial support from Romande Energie, Hoval AG, Holdigaz and Canton of Geneva.

**Conflicts of Interest:** The authors declare no conflict of interest.

## Abbreviations

The following abbreviations are used in this manuscript:

| | |
|---|---|
| DEN | District energy network |
| IDA ICE | IDA indoor climate energy |
| SFH | Single-family house |
| MFH | Multi-family house |
| capex | Capital expenditure |
| SOFC-GT | Solid oxide fuel cell - gas turbine |
| SOEC | Solid oxide electrolysis cell |
| MILP | Mixed integer linear programming |
| PV | Photo-voltaic |
| ERA | Energy reference area |
| DB | Davies - Bouldin (index) |
| ELDC | Error load duration curve |
| GI | Global horizontal irradiation |
| SF | Self-sufficiency |
| SC | Self-consumption |
| BOI | Boiler |
| ELH | Electrical heater |
| AHP | Air-water heat pump |
| COP | Coefficient of performance |
| VAC | Refrigeration cycle |
| SOC | State of charge |
| HST | Heat storage tank |
| HP | Heat pump |
| HE | Heat exchanger |
| BAT | Battery stack |
| DHN | District heating network |

## Appendix A.

*Appendix A.1. Unit Models at Building Level*

Appendix A.1.1. Building

The thermal behaviour of the building is described using a first order dynamic 1R1C model, as illustrated in Figure A1. The entire construction is aggregated into a single capacity $C_b$ while considering a single temperature node $T_b$ [47,48]. Equation (A1) highlights the corresponding energy balance, where $T_b$ denotes the internal temperature, $T^{\text{ext}}$ the external temperature, $U_b^{\text{ext}} = 1/R_b^{\text{ext}}$ the combined thermal transfer coefficient, $\Phi_b^{\text{sun+o}}$ the stochastic gains from solar and occupancy sources and $\dot{Q}_b^+$ the heat supplied by the energy system. In the case of partially non-residential dwellings with cooling requirements, a second zone is added to the model and connected through the internal insulation resistance $R_b^{\text{int}}$. $T^{\text{min/max}}$ in Equation (A2) define the comfort tolerance on the internal temperature.

$$C_b \cdot (T_{b,p,t+1} - T_{b,p,t}) = U_b^{\text{ext}} \cdot (T_{p,t}^{\text{ext}} - T_{b,p,t}) + \Phi_{b,p,t}^{\text{sun+o}} + \dot{Q}_{b,p,t}^+ \qquad \forall p \in \mathbf{P}, \ t \in \mathbf{TOP} \qquad \text{(A1)}$$

$$T_{b,p,t}^{\text{min}} \leq T_{b,p,t} \leq T_{b,p,t}^{\text{max}} \qquad \forall p \in \mathbf{P}, \ t \in \mathbf{TOP} \qquad \text{(A2)}$$

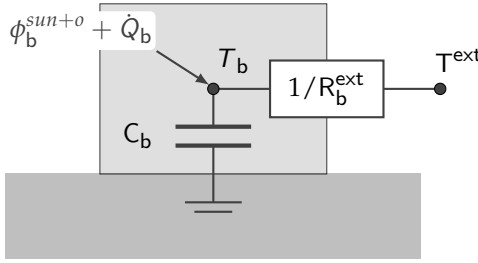

**Figure A1.** 1R1C building model.

Appendix A.1.2. Boiler (BOI)

The natural gas boiler is described using a static system model formulation (Equations (3)–(7)) and is implemented as an auxiliary heating utility, the sizing dimension being the thermal power output. The main parameter required to model this unit, the thermal efficiency ($\epsilon_{BOI}$), is listed in Table A1.

**Table A1.** Parameter data.

| Parameter | Value | Unit | Ref. |
|:---:|:---:|:---:|:---:|
| $\epsilon_{BOI}$ | 0.98 | [-] | [44] |

Appendix A.1.3. Electrical Heater (ELH)

As with the boiler unit, the electrical heater is also described using a static system model formulation (Equations (3)–(7)) and implemented as an auxiliary heating utility, the sizing dimension being the thermal power output. The main parameter required to model this unit, the thermal efficiency ($\epsilon_{ELH}$), is listed in Table A2.

**Table A2.** Parameter data (ELH).

| Parameter | Value | Unit | Ref. |
|:---:|:---:|:---:|:---:|
| $\epsilon_{ELH}$ | 0.98 | [-] | estimate |

Appendix A.1.4. Heat Pumps

The air-source heat pump unit (AHP) is described using a static system formulation (Equations (3)–(7)), the sizing dimension being the electrical power input. The conversion efficiency (Equations (A3) and (A4)) is determined using the ideal coefficient of performance (COP) and the second law efficiency $\eta$, which accounts for the irreversibilities in the different cycle components (e.g., compressor). In order to avoid non-linearities coming from the variable supply temperature, the generated heat load is discretised into $n_s = |\mathbf{S}_{AHP}|$ streams $s$. When considering different heat sources (e.g., water-source heat pumps) in the problem formulation, a similar model definition can be applied, the solely modification being the source temperature (e.g., $T_{p,t}^{water}$) and the respective second-law efficiency $\eta$. The values of the parameters considered for the air-water heat pump are given in Table A3 and for the corresponding refrigeration cycle (VAC) in Table A4.

$$\text{COP}_{AHP,s,p,t} = \frac{T_{AHP,s}^{sink}}{T_{AHP,s}^{sink} - T_{p,t}^{source}} \qquad \forall s \in \mathbf{S}_{AHP},\ p \in \mathbf{P},\ t \in \mathbf{TOP} \qquad (A3)$$

$$\dot{q}_{AHP,s,p,t}^{-} = \eta_{AHP,s,p,t} \cdot \text{COP}_{AHP,s,p,t} \qquad \forall s \in \mathbf{S}_{AHP},\ p \in \mathbf{P},\ t \in \mathbf{TOP} \qquad (A4)$$

**Table A3.** Default parameters values for the AHP second-law efficiency and part-load limit, evaluated from [49].

| | Par. | $T^{sink}$ | $T^{source}$ [°C] | | | | | | | | | |
|---|---|---|---|---|---|---|---|---|---|---|---|---|
| | | [°C] | −20 | −15 | −10 | −7 | −2 | 2 | 7 | 10 | 15 | 20 |
| **AHP** | $\eta$ | 35 | 0 | 0.464 | 0.458 | 0.458 | 0.469 | 0.462 | 0.435 | 0.416 | 0.37 | 0.307 |
| | | 45 | 0 | 0.445 | 0.463 | 0.464 | 0.46 | 0.446 | 0.439 | 0.436 | 0.43 | 0.396 |
| | | 55 | 0 | 0 | 0 | 0.421 | 0.423 | 0.416 | 0.439 | 0.436 | 0.412 | 0.395 |
| | $\dot{m}^{+,max}_{electricity}$ | 35 | 0 | 0.62 | 0.65 | 0.65 | 0.65 | 0.65 | 0.68 | 0.68 | 0.68 | 0.68 |
| | | 45 | 0 | 0.74 | 0.74 | 0.74 | 0.76 | 0.79 | 0.82 | 0.82 | 0.79 | 0.79 |
| | | 55 | 0 | 0 | 0 | 0.91 | 0.94 | 0.97 | 0.97 | 0.97 | 1 | 1 |

**Table A4.** Default parameters values for the VAC second-law efficiency and part-load limit, evaluated from [49].

| | Par. | $T^{sink}$ | $T^{source}$ [°C] | | | | | |
|---|---|---|---|---|---|---|---|---|
| | | | 20 | 25 | 30 | 35 | 40 | 45 |
| **VAC** | $\eta$ | 13 | 0.103 | 0.159 | 0.198 | 0.219 | 0.249 | 0.224 |
| | | 15 | 0.076 | 0.14 | 0.181 | 0.243 | 0.243 | 0.224 |
| | | 18 | 0.033 | 0.101 | 0.146 | 0.209 | 0.209 | 0.218 |
| | | 22 | 0 | 0.005 | 0.106 | 0.184 | 0.184 | 0.215 |
| | $\dot{m}^{+,max}_{electricity}$ | 13 | 0.71 | 0.78 | 0.86 | 0.95 | 0.91 | 1 |
| | | 15 | 0.73 | 0.78 | 0.87 | 0.95 | 0.91 | 0.96 |
| | | 18 | 0.73 | 0.8 | 0.89 | 0.96 | 0.93 | 0.89 |
| | | 22 | 0.75 | 0.82 | 0.91 | 1 | 0.95 | 0.8 |

Appendix A.1.5. Storage Units

Battery Stack (BAT)

Stationary batteries are described using a single state dynamic model, the sizing dimension being the electrical energy stored. The model accounts for the system self-discharging rate ($\sigma$) as well as the charging and discharging losses ($\gamma$). To limit any premature degradation of the stack, the minimum ($SOC^{min}_{BAT}$) and maximum ($SOC^{max}_{BAT}$) battery states of charge (SOC) are fixed (Equations (A5) and (A6)). The parameters used to model this unit are listed in Table A5.

$$f_{BAT,p,t} \geq SOC^{min}_{BAT} \cdot f_{BAT} \qquad \forall p \in \mathbf{P},\ t \in \mathbf{TOP} \tag{A5}$$

$$f_{BAT,p,t} \leq SOC^{max}_{BAT} \cdot f_{BAT} \qquad \forall p \in \mathbf{P},\ t \in \mathbf{TOP} \tag{A6}$$

**Table A5.** Parameter data (BAT).

| Parameter | Value | Unit | Ref. |
|---|---|---|---|
| $\gamma^{ch}$ | 0.9 | [-] | [39] |
| $\gamma^{dch}$ | 0.9 | [-] | [39] |
| $\sigma$ | 0 | [-] | [39] |
| $SOC^{max}_{BAT}$ | 0.8 | [-] | [50] |
| $SOC^{min}_{BAT}$ | 0.2 | [-] | [50] |

Heat Storage Tanks (HST)

The thermal energy storage tanks are described through a single state, first order dynamic model formulation, the sizing dimension being the unit volume. The minimum state of charge $SOC^{min}$ is set as the current building return temperature $T^{h,r}_{b,p,t}$ during space heating periods, while the maximum

operating temperature $T_{HST}^{max}$ is defined as the lowest value between the heat pump operating limit and the nominal supply temperature of the heating system ($T_b^{h,s}$). The required parameters include the tank diameter $D_{HST}$, the specific heat loss rate $U_{HST}$ as well as the charging and discharging efficiencies $\gamma$. The unit is consequently added into the heat cascade formulation through the single charging (cold) and discharging (hot) streams as defined in Equations (A7)–(A10). The parameter values are given in Table A6.

$$\sigma_{HST} = \frac{4 \cdot U_{HST}}{D_{HST}} \cdot \left( T_{HST}^{max} - T_{b,p,t}^{h,r} \right) \qquad \forall p \in \mathbf{P}, \ t \in \mathbf{TOP} \tag{A7}$$

$$\kappa_{HST} = \frac{4 \cdot U_{HST}}{D_{HST}} \cdot \left( T_{b,p,t}^{h,r} - T^{amb} \right) \qquad \forall p \in \mathbf{P}, \ t \in \mathbf{TOP} \tag{A8}$$

$$\dot{q}_{HST,s,p,t}^{+} = c_p \cdot \rho \cdot \left( T_{HST}^{max} - T_{b,p,t}^{h,r} \right) \qquad \forall p \in \mathbf{P}, \ t \in \mathbf{TOP} \tag{A9}$$

$$\dot{q}_{HST,s,p,t}^{-} = c_p \cdot \rho \cdot \left( T_{HST}^{max} - T_{b,p,t}^{h,r} \right) \qquad \forall p \in \mathbf{P}, \ t \in \mathbf{TOP} \tag{A10}$$

**Table A6.** Parameter data (HST).

| Parameter | Value | Unit | Ref. |
|---|---|---|---|
| $c_p$ | 4.186 | [kJ/(kg $\cdot$K)] | estimate |
| $\rho$ | 1000 | [kg/m$^3$] | estimate |
| $\gamma^{ch}$ | 0.99 | [-] | estimate |
| $\gamma^{dch}$ | 0.99 | [-] | estimate |
| $D_{HST}$ | 0.98 | [m] | estimate |
| $U_{HST}$ | 0.0013 | [kW/m$^2$] | [44] |

*Appendix A.2. Unit Models at City Level*

Appendix A.2.1. PV Panels

The PV panels are modeled as described in [47], with $A^{PV}$ the PV area, $\eta^{PV}$ the PV efficiency, $I^{sun}$ the irradiation of the sun, $T^{PV}$ the PV temperature, $U^{glass}$ the thermal transmission coefficient, $T^{amb}$ the ambient temperature, and $f^{glass}$ the factor denoting the portion of the solar irradiation passing through the PV glass:

$$\dot{m}_{PV,electricity}^{-} = A^{PV} \cdot \eta^{PV} \cdot I^{sun} \tag{A11a}$$

$$\eta^{PV} = \eta^{PV,ref} - \eta^{PV,var} \cdot (T^{PV} - T^{PV,ref}) \tag{A11b}$$

$$T^{PV} = \frac{U^{glass} \cdot T^{amb}}{U^{glass} - \eta^{PV,var} \cdot I^{sun}} + \frac{I^{sun} \cdot (f^{glass} - \eta^{PV,ref} - \eta^{PV,var} \cdot T^{PV,ref})}{U^{glass} - \eta^{PV,var} \cdot I^{sun}} \tag{A11c}$$

The different parameters assumed are given in Table A7 [47] and the reference stream for $A^{PV} = 100 \ m^2$ and $I^{sun} = 100 \ W/m^2$ is given in Table A8.

**Table A7.** Parameters for PV panels.

| Parameter | Value | Unit |
|---|---|---|
| $T^{amb}$ | 288 | K |
| $T^{PV,ref}$ | 298 | K |
| $U^{glass}$ | 29.1 | W/(m$^2\cdot$K) |
| $f^{glass}$ | 0.9 | - |
| $\eta^{PV,ref}$ | 0.14 | - |
| $\eta^{PV,var}$ | 0.001 | $1/K$ |

**Table A8.** Streams for PV panel.

| Type | $T^{in}$ [°C] | $T^{out}$ [°C] | $\dot{Q}$ | $\dot{m}^-$ | $\dot{m}^+$ |
|---|---|---|---|---|---|
| Electricity | - | - | - | 1.66 kW | - |

Appendix A.2.2. SOEC-SOFC Co-Generation and Methanation

The co-generation SOFC-GT unit is modeled according to [32] and the co-generation SOEC unit according to [33]. A list of the reference streams in the different units are given in Tables A9–A11.

**Table A9.** Streams for SOEC unit.

| Type | $T^{in}$ [°C] | $T^{out}$ [°C] | $\dot{Q}$ | $\dot{m}^-$ | $\dot{m}^+$ |
|---|---|---|---|---|---|
| Heat | 91 | 58 | 3.05 kW | - | - |
| Heat | 58 | 27 | 1.66 kW | - | - |
| Electricity | - | - | - | - | 100 kW |
| $H_2O$ | - | - | - | - | 5.98 g/s |
| $H_2$ | - | - | - | 0.67 g/s (94.21 kW) | - |

**Table A10.** Streams for SOFC-GT unit.

| Type | $T^{in}$ [°C] | $T^{out}$ [°C] | $\dot{Q}$ | $\dot{m}^-$ | $\dot{m}^+$ |
|---|---|---|---|---|---|
| Heat | 648.8 | 260.0 | 16.28 kW | - | - |
| Heat | 109.8 | 35.2 | 9.44 kW | - | - |
| Heat | 35.2 | 30.2 | 1.44 kW | - | - |
| Electricity | - | - | - | 100 kW | - |
| $CH_4$ | - | - | - | - | -2.41 g/s (133.48 kW) |
| $CO_2$ | - | - | - | 6.60 g/s | - |

**Table A11.** Streams for methanation unit.

| Type | $T^{in}$ [°C] | $T^{out}$ [°C] | $\dot{Q}$ | $\dot{m}^-$ | $\dot{m}^+$ |
|---|---|---|---|---|---|
| Heat | 625.4 | 507.3 | 138.4 kW | - | - |
| Heat | 507.3 | 507.1 | 0.3 kW | - | - |
| Heat | 507.1 | 233.0 | 585.3 kW | - | - |
| Heat | 233.0 | 228.0 | 9.3 kW | - | - |
| Heat | 228.0 | 227.0 | 0.7 kW | - | - |
| Heat | 227.0 | 215.0 | 12.7 kW | - | - |
| Heat | 215.0 | 203.0 | 27.1 kW | - | - |
| Heat | 203.0 | 186.7 | 25.3 kW | - | - |
| Heat | 186.7 | 28.0 | 358.0 kW | - | - |
| Electricity | - | - | - | 100 kW | - |
| $H_2$ | - | - | - | - | 0.2 kg/s (28,349.2 kW) |
| $CO_2$ | - | - | - | - | 1.1 kg/s |
| $CH_4$ | - | - | - | 0.4 kg/s (22,193.6 kW) | - |

The reference flows for the SOEC unit are given for an incoming flow of electricity of 100 kW. The electricity to hydrogen efficiency is computed using the HHV of $H_2$ of 141,746 kJ/kg [51]:

$$\eta = \frac{\dot{m}^-_{H_2} \cdot \text{HHV}^{H_2}}{\dot{m}^+_{electricity}} = 94.2\% \tag{A12}$$

The reference flows for the SOFC-GT unit are given for an outgoing flow of electricity of 100 kW. The electrical and thermal efficiencies are calculated using the HHV of $CH_4$ of 55,484 kJ/kg [51]:

$$\eta^{el} = \frac{\dot{m}^-_{electricity}}{\dot{m}^+_{CH_4} \cdot HHV^{CH_4}} = 74.9\% \tag{A13a}$$

$$\eta^{th} = \frac{\sum_{k \in \mathbf{K}} \dot{Q}_{SOFC}}{\dot{m}^+_{CH_4} \cdot HHV^{CH_4}} = 20.3\% \tag{A13b}$$

The reference flows for the metahantion unit are given for an incoming flow of electricity of 100 kW.

Appendix A.2.3. Steam Network

In the steam network, steam is produced at very high pressure and distributed at multiple lower pressure levels. The pressure levels are selected to fit the production profiles of the P2G units. The parameters used to model the steam network are summarized in Table A12.

**Table A12.** Parameters for steam network.

| Type | Header Pressure [bar] | $T^{superheat}$ [°C] | Turbine |
|------|----------------------|----------------------|---------|
| Production | 120 | 100 | yes |
| Distribution | 30 | 2 | yes |
| Distribution | 10 | 2 | yes |
| Distribution | 5 | 2 | yes |
| Distribution | 2 | 2 | no |
| Distribution | 1 | 2 | no |
| Distribution | 0.2 | 2 | no |

Appendix A.2.4. $CO_2$ and $CH_4$ Storage

The storage tanks are modeled using the following equations:

$$SL_{tank,t+1} = SL_{tank,t} + \eta^{ch} \cdot \dot{M}^+_{fuel,t} - \frac{1}{\eta^{dch}} \cdot \dot{M}^-_{fuel,t} \tag{A14a}$$

$$SL_{tank,t} = f_{tank,t} \qquad \forall t \in \mathbf{T} \tag{A14b}$$

where $SL_{tank,t}$ represents the storage level of the tank at time step $t$, $\dot{M}^+_{fuel,t}$ and $\dot{M}^-_{fuel,t}$ the flow rates in and out of the unit at time step $t$, and $\eta^{ch}, \eta^{dch}$ the charging and discharging efficiencies. $CO_2$ is stored in liquid form at atmospheric pressure and temperature (i.e., 1 bar, 25 °C). Methane is also stored as a liquid, at the operating pressure of 1 bar and the corresponding temperature required for the liquid state, of $-162$ °C.

Appendix A.2.5. Central Plants

The central plant in winter is modeled as a HP using a lake (at a constant temperature of 7.5 °C) as the heat source and $CO_2$ as the refrigerant. A summary of the parameters used for the central plant HP can be observed in table A13.

**Table A13.** Parameters for central plant HP.

| Unit | HP Central Plant |
|------|------------------|
| $T^{subcool}$ [°C] | 1 |
| $T^{superheat}$ [°C] | 2 |
| $\eta^{comp}$ [−] | 0.8 |
| $dT^{min,\,evap}$ [°C] | 5.5 |
| $dT^{min,\,cond}$ [°C] | 1 |

The reference flow of the central plant HP is $\dot{Q}^{cond} = \dot{m}_{CO_2} \cdot L^v_{CO_2}$ and the electricity consumption of the compressor and the heat extracted at the evaporator are calculated solving the thermodynamic cycle. The reference streams of the unit are given in Table A14 for a mass flow of $CO_2$ of 1 kg/s. The COP of the central plant HP is constant throughout the year, at 15.1.

**Table A14.** Streams for central plant HP (winter).

| Type | $T^{in}$ [°C] | $T^{out}$ [°C] | $\dot{Q}$ | $\dot{m}^-$ | $\dot{m}^+$ |
|---|---|---|---|---|---|
| Heat evaporator | 2 | 4 | $186.4 \cdot \frac{COP-1}{COP}$ kW | - | - |
| Heat condensor | 15 | 13 | 186.4 kW | - | - |
| Electricity | - | - | - | - | $186.4 \cdot \frac{1}{COP}$ kW |
| $CO_2{}^{vap}$ | - | - | - | 1 kg/s | - |
| $CO_2{}^{liq}$ | - | - | - | - | 1 kg/s |

The central plant in summer is modeled as a HE with the reference flow $\dot{Q} = \dot{m}_{CO_2} \cdot L^v_{CO_2}$ and a minimum temperature difference $dT_{min}$ = 5.5°C. The reference streams of the unit are given in Table A15 for a mass flow of $CO_2$ of 1 kg/s.

**Table A15.** Streams for central plant HE (summer).

| Type | $T^{in}$ [°C] | $T^{out}$ [°C] | $\dot{Q}$ | $\dot{m}^-$ | $\dot{m}^+$ |
|---|---|---|---|---|---|
| Heat | 7.5 | 9.5 | 186.4 kW | - | - |
| $CO_2{}^{vap}$ | - | - | - | - | 1 kg/s |
| $CO_2{}^{liq}$ | - | - | - | 1 kg/s | - |

Appendix A.2.6. Investment Cost of Energy Conversion Technologies

The fixed and variable IC parameters, as well as the reference flows for the different units can be found in Table A16.

**Table A16.** Parameters for IC.

| Unit | $C^{inv,1}$ [€] | $C^{inv,2}$ [€/kW/€/m$^2$] | Attribute |
|---|---|---|---|
| Boiler | 3990 | 110 | $\dot{Q}^-$ [kW] |
| Electrical heater | 968 | 13 | $\dot{Q}^-$ [kW] |
| Heat pumps/Ref cycle | 10,224 | 2232 | $\dot{m}^+_{electricity}$ [kW] |
| Battery stack | 825 | 1290 | $\max(f_{BAT,p,t})$ [kW] |
| Heat storage tank | 1421 | 1945 | V [kW] |
| Domestic hot water tank | 496 | 10,248 | V [kW] |
| PV panels | - | 247 | $A^{PV}$ [m$^2$] |
| SOEC-SOFC | - | 4760 | $\max(\dot{m}^+_{electricity, SOEC}, \dot{m}^+_{electricity, SOFC})$ [kW] |
| HP CP (winter) | 5680 | 1240 | $\dot{m}^+_{electricity}$ [kW] |
| HE CP (summer) | 184 | 197 | $A^{HE}$ [m$^2$] |

*Appendix A.3. Heat Distribution Cost*

The heat distribution cost of the networks is calculated using the formulation of [31]. First, the length of the network ($L^{DHN}$) is calculated based on the number of buildings ($n_b$), the land surface area ($A_l$) and a correlation coefficient (K) [31]:

$$L^{DHN} = 2 \cdot (n_b - 1) \cdot K \cdot \sqrt{\frac{A_l}{n_b}} \tag{A15}$$

And for each segment (between each two buildings):

$$L_k^{DHN} = \frac{L^{DHN}}{n_b}$$

(A16)

Next, the mass flow in the pipes is computed using the maximum heat flow in the pipe $\dot{Q}^{DHN}$ and the specific heat flows $q_{water} = c_{p,water}(T_{supply} - T_{return})$, $q_{CO_2} = l_v$:

$$\dot{m}_{max}^{DHN} = \frac{\dot{Q}^{DHN}}{q^{DHN}}$$

(A17)

And for each segment ($k$):

$$\dot{m}_k^{DHN} = \frac{\dot{Q}^{DHN} \cdot (n_b - k + 1)}{n_b \cdot q^{DHN}}$$

(A18)

Then, the diameter of the pipes ($d^{DHN}$) is calculated using the mass flow $\dot{m}^{DHN}$, the sizing velocity of the fluids (v) [52] and the density of the fluids ($\rho$):

$$d_k^{DHN} = \sqrt{\frac{4 \cdot \dot{m}_k^{DHN}}{\pi \cdot v \cdot \rho}}$$

(A19)

Finally, the investment cost ($C^{inv}$) of the networks is computed by summing up the different segments, using the cost coefficients $c_1$ and $c_2$ [52], an interest rate i = 5% and a lifetime lt = 60 years [31]:

$$\tau^{DHN} = \frac{(i+1)^{lt} - 1}{i \cdot (i+1)^{lt}}$$

(A20)

$$C^{inv} = \sum_{k=1}^{n-1} \frac{L_k^{DHN}(c_1 \cdot d_k^{DHN} + c_2)}{\tau^{DHN}}$$

(A21)

The values of the parameters present in the equations above can be found in Table A17.

**Table A17.** Network cost parameters.

| Parameter | Unit | Value ($CO_2$ Network) | Value ($H_2O$ Network) |
|---|---|---|---|
| $n_b$ | [-] | 11,903 | 11,903 |
| K | [-] | 0.23 | 0.23 |
| $A_l$ | [m$^2$] | 15,785,286 | 15,785,286 |
| $L^{DHN}$ | [km] | 3630.3 | 3630.3 |
| $q^{DHN}$ | [kJ/kg] | 186.4 | 18.8 |
| $\dot{Q}^{DHN}$ | [MW] | 2938.1 | 2942.7 |
| $\dot{m}_{max}^{DHN}$ | [t/s] | 15.8 | 156.5 |
| v | [m$^2$/s] | 3 (liquid), 6 (vapour) | 3 |
| $\rho$ | [kg/m$^3$] | 837.7 (liquid), 160.9 (vapour) | 1000 |
| $d_{max}^{DHN}$ | [m] | 4 (liquid), 10.4 (vapour) | 33.2 |
| $c_1$ | [€/m$^2$] | 5670 | 5670 |
| $c_2$ | [€] | 613 | 613 |
| i | [-] | 0.06 | 0.06 |
| lt | [-] | 60 | 60 |
| $C^{inv}$ | [M€/y] | 153.5 | 330.6 |

*Appendix A.4. RegBL Database Parameter Names*

**Table A18.** RegBL database corresponding parameter notations.

| Parameter Description | Notation (This Paper) | Notation (RegBL) |
|---|---|---|
| Building category | - | GKAT |
| Building class | - | GKLAS |
| Building footprint area | $A_b$ | GAREA |
| Building number of floors | $N^{floors}$ | GASTW |
| Building rooftop area | $A_b^{roof}$ | FLAECHE |
| Building average solar irradiation | $I_b$ | MSTRAHLUNG |

*Appendix A.5. Energy Service Demand of Different Building Categories*

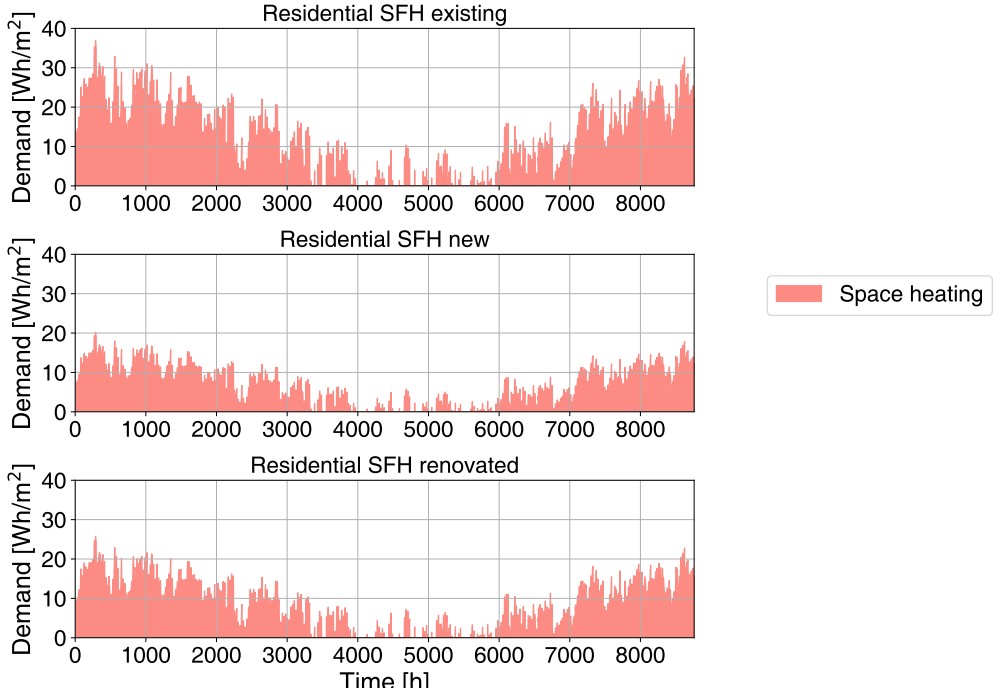

**Figure A2.** Hourly specific energy service demand of residential SFH buildings.

**Table A19.** Yearly specific energy service demand of residential SFH buildings.

| Building Renovation Stage | Space Heating [kWh/m²] | Air Cooling [kWh/m²] | Dom. Hot Water [kWh/m²] | Electricity [kWh/m²] |
|---|---|---|---|---|
| Existing | 80.3 | 0.0 | 13.6 | 18.2 |
| New | 44.0 | 0.0 | 13.6 | 18.2 |
| Renovated | 55.9 | 0.0 | 13.6 | 18.2 |

**Table A20.** Yearly specific energy service demand of residential MFH buildings.

| Building Renovation Stage | Space Heating [kWh/m²] | Air Cooling [kWh/m²] | Dom. Hot Water [kWh/m²] | Electricity [kWh/m²] |
|---|---|---|---|---|
| Existing | 80.3 | 0.0 | 17.8 | 18.4 |
| New | 44.0 | 0.0 | 17.8 | 18.4 |
| Renovated | 55.9 | 0.0 | 17.8 | 18.4 |

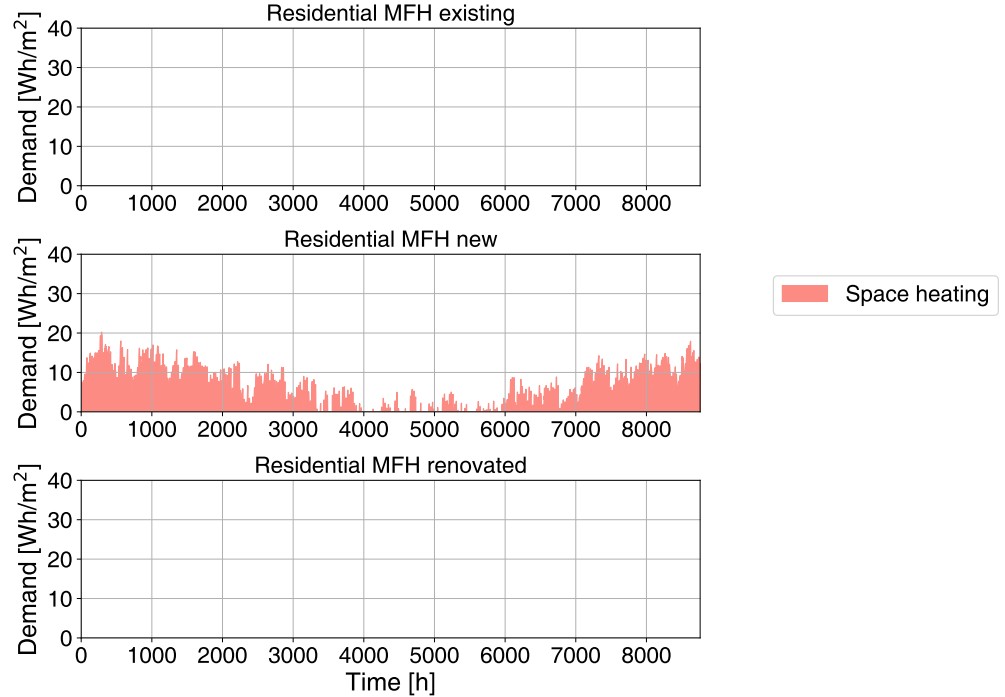

**Figure A3.** Hourly specific energy service demand of residential MFH buildings.

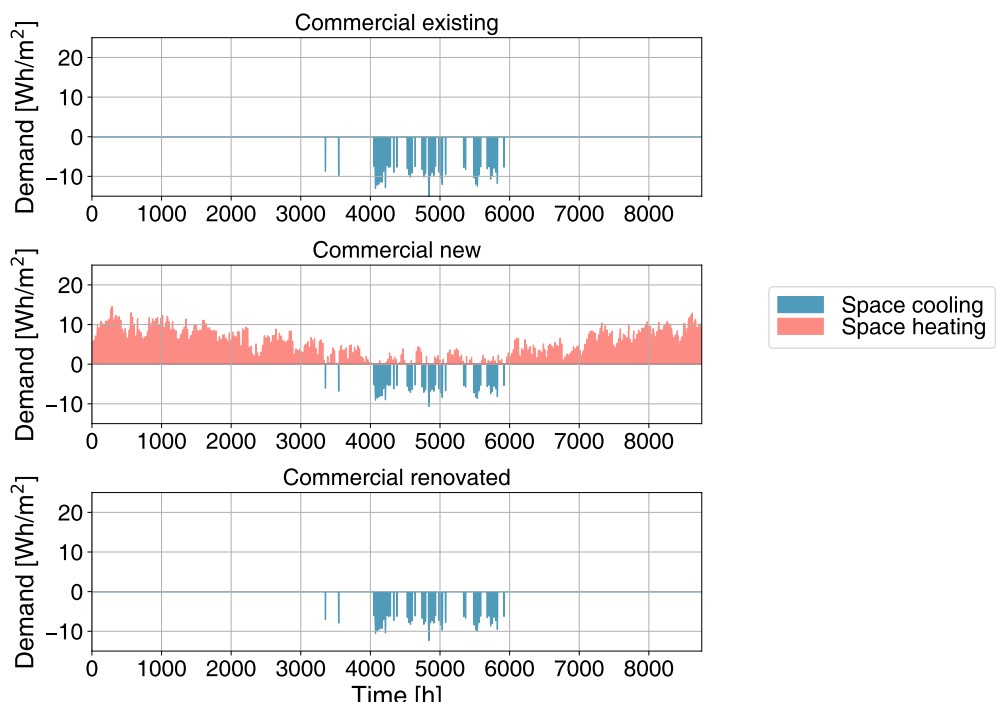

**Figure A4.** Hourly specific energy service demand of commercial buildings.

**Table A21.** Yearly specific energy service demand of commercial buildings.

| Building Renovation Stage | Space Heating [kWh/m²] | Air Cooling [kWh/m²] | Dom. Hot Water [kWh/m²] | Electricity [kWh/m²] |
|---|---|---|---|---|
| Existing | 49.2 | 3.3 | 1.8 | 114.4 |
| New | 33.5 | 2.3 | 1.8 | 114.4 |
| Renovated | 38.4 | 2.7 | 1.8 | 114.4 |

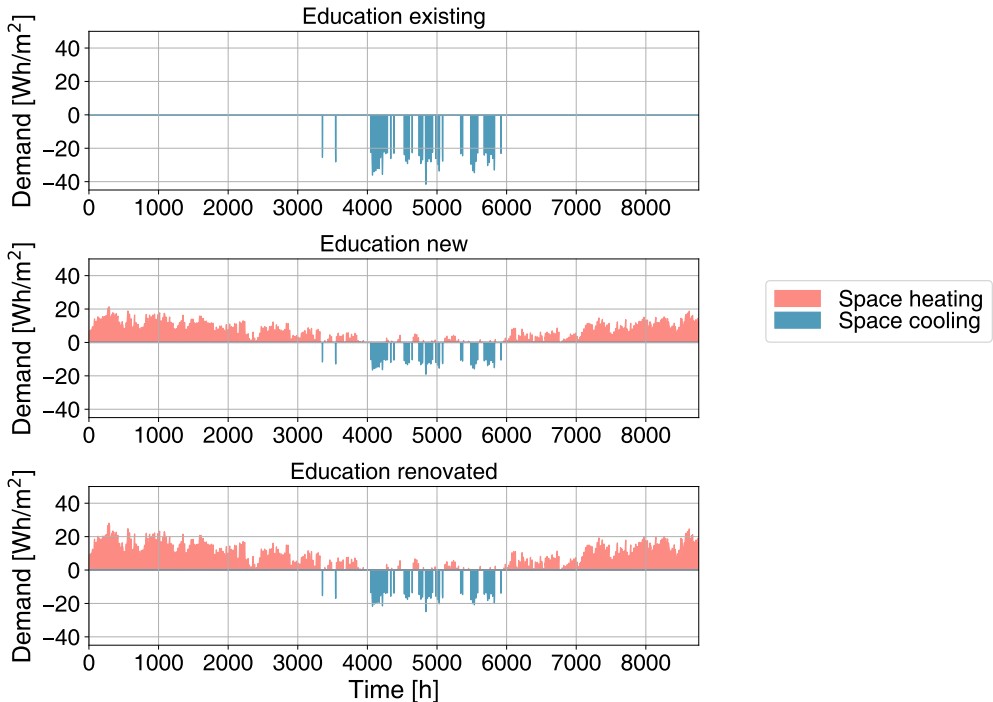

**Figure A5.** Hourly specific energy service demand of education buildings.

**Table A22.** Yearly specific energy service demand of education buildings.

| Building Renovation Stage | Space Heating [kWh/m$^2$] | Air Cooling [kWh/m$^2$] | Dom. Hot Water [kWh/m$^2$] | Electricity [kWh/m$^2$] |
|---|---|---|---|---|
| Existing | 91.8 | 9.7 | 4.5 | 23.8 |
| New | 41.9 | 4.4 | 4.5 | 23.8 |
| Renovated | 55.1 | 5.8 | 4.5 | 23.8 |

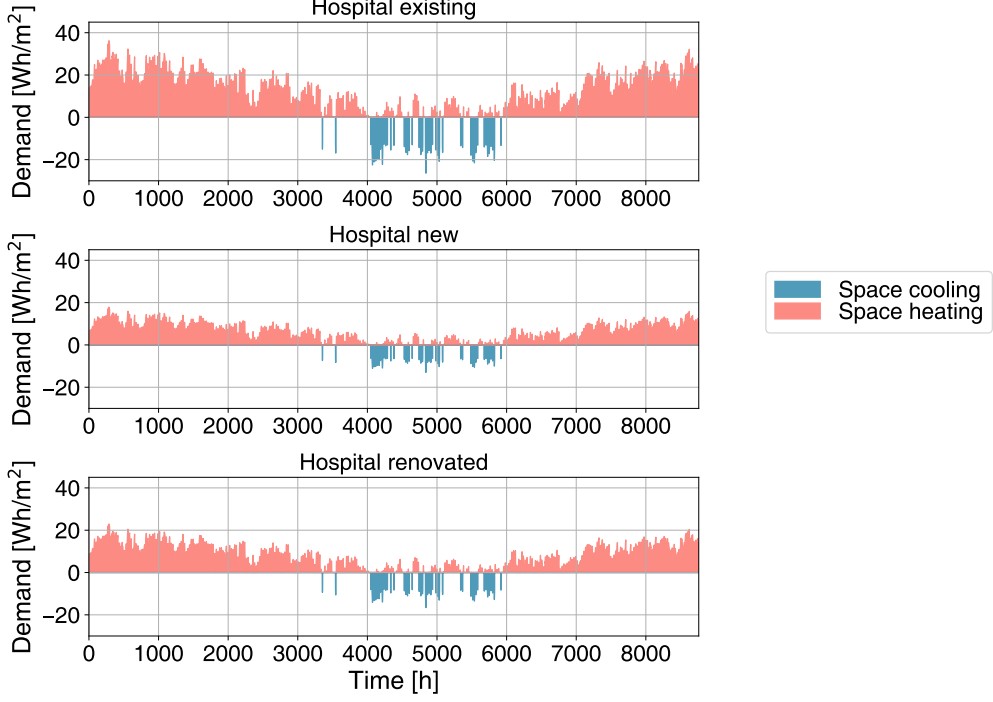

**Figure A6.** Hourly specific energy service demand of hospital buildings.

**Table A23.** Yearly specific energy service demand of hospital buildings.

| Building Renovation Stage | Space Heating [kWh/m$^2$] | Air Cooling [kWh/m$^2$] | Dom. Hot Water [kWh/m$^2$] | Electricity [kWh/m$^2$] |
|---|---|---|---|---|
| Existing | 83.5 | 5.8 | 34.1 | 34.0 |
| New | 41.2 | 2.8 | 34.1 | 34.0 |
| Renovated | 53.3 | 3.6 | 34.1 | 34.0 |

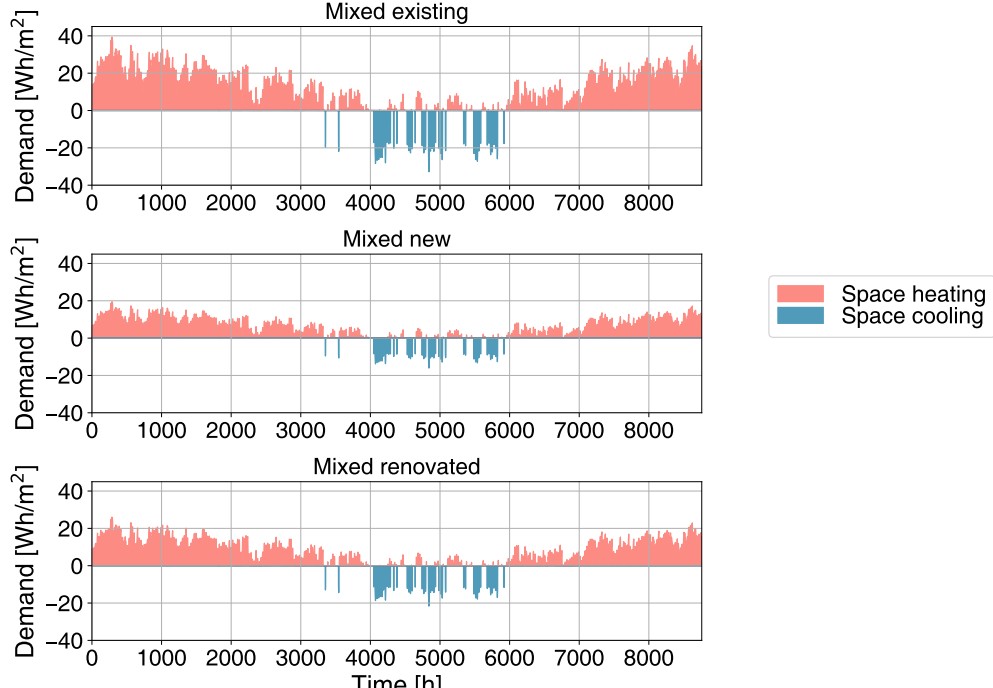

**Figure A7.** Hourly specific energy service demand of mixed buildings.

**Table A24.** Yearly specific energy service demand of mixed buildings.

| Building Renovation Stage | Space Heating [kWh/m$^2$] | Air Cooling [kWh/m$^2$] | Dom. Hot Water [kWh/m$^2$] | Electricity [kWh/m$^2$] |
|---|---|---|---|---|
| Existing | 81.6 | 7.5 | 11.9 | 28.4 |
| New | 40.9 | 3.7 | 11.9 | 28.4 |
| Renovated | 54.1 | 4.9 | 11.9 | 28.4 |

**Table A25.** Heat signature coefficients for all building types and ages.

| Building Type | Building Age | k1 [kW/(m²·°C)] | k2 [kW/m²] | T$_{base, h}$ [°C] | T$_{base, c}$ [°C] |
|---|---|---|---|---|---|
| Residential SFH | existing | −1.52 | 23.59 | 15.52 | - |
| | new | −0.83 | 12.91 | 15.55 | - |
| | renovated | −1.06 | 16.43 | 15.5 | - |
| Residential MFH | existing | −1.52 | 23.59 | 15.52 | - |
| | new | −0.83 | 12.91 | 15.55 | - |
| | renovated | −1.06 | 16.43 | 15.5 | - |
| Administrative | existing | −1.87 | 26.51 | 14.18 | 25 |
| | new | −0.8 | 11.41 | 14.26 | 25 |
| | renovated | −1.15 | 16.29 | 14.17 | 25 |
| Commercial | existing | −0.84 | 13.81 | 16.44 | 25 |
| | new | −0.58 | 9.47 | 16.33 | 25 |
| | renovated | −0.67 | 10.89 | 16.25 | 25 |
| Education | existing | −2.03 | 28.84 | 14.21 | 25 |
| | new | −0.93 | 13.19 | 14.18 | 25 |
| | renovated | −1.22 | 17.32 | 14.2 | 25 |
| Hospital | existing | −1.44 | 23.54 | 16.34 | 25 |
| | new | −0.71 | 11.62 | 16.37 | 25 |
| | renovated | −0.91 | 14.96 | 16.44 | 25 |
| Mixed | existing | −1.86 | 27.83 | 14.98 | 25 |
| | new | −0.82 | 12.31 | 15.05 | 25 |
| | renovated | −1.1 | 16.37 | 14.94 | 25 |

*Appendix A.6. Results from Parallel Coordinates*

This section presents the detailed results presented schematically in Figures 16 and 17.

**Table A26.** Detailed results for Figure 16.

| Investment Cost M€/year | Operating Cost M€/year | CO$_2$ Emissions kt$_{CO_2}$/year) | Buildings High IC - | Buildings Low IC - | CO$_2$ Activation - | Self Sufficiency % | PV Market M€/year | HP Market M€/year |
|---|---|---|---|---|---|---|---|---|
| 1.61 | 1.49 | 3.06 | 0 | 46,121 | 0 | 0 | 0.00 | 0.55 |
| 5.53 | 0.84 | 1.77 | 604 | 45,517 | 1 | 0 | 1.78 | 1.62 |
| 9.44 | 0.56 | 1.13 | 10,164 | 35,957 | 2 | 0 | 5.36 | 2.40 |
| 12.8 | 0.44 | 0.92 | 9088 | 37,033 | 3 | 24 | 7.61 | 2.39 |
| 14.03 | 0.35 | 0.76 | 8696 | 37,425 | 4 | 62 | 7.97 | 2.37 |
| 15.50 | 0.26 | 0.62 | 7826 | 38,293 | 6 | 89 | 7.96 | 2.35 |
| 17.56 | 0.20 | 0.50 | 7091 | 39,030 | 11 | 100 | 8.24 | 2.32 |
| 19.05 | 0.16 | 0.41 | 7143 | 38,978 | 16 | 100 | 8.09 | 2.29 |
| 20.86 | 0.13 | 0.34 | 5173 | 40,948 | 24 | 100 | 7.78 | 2.29 |
| 23.96 | 0.12 | 0.31 | 3816 | 42,305 | 48 | 100 | 7.76 | 2.21 |

**Table A27.** Detailed results for Figure 17.

| Population Density cap/km² | Building Density Buildings/km² | Network Cost k€/(100 m²) | El Imports GWh/year | El Exports GWh/year | NG Imports GWh/year | Investment Cost €/(100 m² year) | PV Market €/(100 m² year) | HP Market €/(100 m² year) |
|---|---|---|---|---|---|---|---|---|
| 40.11 | 10.15 | 27.33 | 2.39 | 3.40 | 0.45 | 21.50 | 17.55 | 4.79 |
| 58.50 | 17.00 | 35.00 | 2.31 | 3.09 | 0.56 | 21.38 | 17.14 | 4.94 |
| 113.57 | 24.25 | 25.83 | 1.76 | 2.96 | 0.33 | 15.65 | 13.06 | 3.23 |
| 116.61 | 31.49 | 23.22 | 1.71 | 2.48 | 0.33 | 16.40 | 13.35 | 3.72 |
| 160.73 | 46.59 | 26.20 | 2.12 | 3.40 | 0.33 | 16.25 | 13.34 | 3.50 |
| 180.15 | 36.63 | 21.53 | 4.70 | 6.01 | 0.89 | 12.60 | 10.29 | 2.77 |
| 187.03 | 31.44 | 34.56 | 3.72 | 5.14 | 0.67 | 19.49 | 16.4 | 4.09 |
| 196.87 | 62.05 | 26.64 | 3.11 | 4.74 | 0.56 | 17.69 | 14.50 | 3.87 |
| 199.46 | 63.62 | 18.84 | 3.17 | 2.97 | 0.89 | 11.77 | 9.24 | 2.89 |
| 214.65 | 57.61 | 29.01 | 3.15 | 6.20 | 0.45 | 19.35 | 16.36 | 3.79 |
| 260.72 | 63.40 | 24.15 | 4.29 | 5.89 | 1.03 | 11.89 | 9.63 | 2.65 |
| 276.87 | 84.11 | 34.66 | 3.96 | 7.11 | 0.67 | 20.66 | 16.96 | 4.50 |
| 291.60 | 76.49 | 39.68 | 4.48 | 11.13 | 0.33 | 19.03 | 15.89 | 3.98 |
| 309.17 | 108.03 | 49.88 | 3.31 | 5.86 | 0.56 | 29.96 | 24.60 | 6.55 |
| 386.72 | 60.20 | 19.02 | 4.56 | 5.63 | 1.19 | 16.54 | 13.34 | 3.62 |
| 396.37 | 108.66 | 12.55 | 4.02 | 5.36 | 1.00 | 10.33 | 8.34 | 2.30 |
| 426.46 | 108.21 | 24.20 | 4.96 | 7.47 | 0.78 | 19.21 | 115.79 | 4.14 |
| 449.19 | 77.06 | 18.32 | 5.62 | 7.31 | 1.33 | 13.30 | 11.00 | 2.77 |
| 451.83 | 69.68 | 12.20 | 4.36 | 6.58 | 1.74 | 13.84 | 11.05 | 2.95 |
| 453.07 | 93.66 | 19.60 | 8.45 | 8.57 | 2.00 | 15.85 | 12.65 | 3.71 |
| 516.24 | 104.00 | 24.50 | 13.50 | 22.37 | 2.80 | 14.61 | 12.30 | 2.90 |
| 522.15 | 108.09 | 24.30 | 9.19 | 11.76 | 2.11 | 15.77 | 12.72 | 3.60 |
| 523.17 | 158.29 | 27.67 | 7.18 | 13.80 | 1.25 | 13.60 | 19.49 | 5.04 |
| 610.19 | 109.39 | 25.75 | 8.43 | 15.36 | 0.94 | 17.77 | 14.90 | 3.61 |
| 612.01 | 153.92 | 18.14 | 7.70 | 9.80 | 1.92 | 15.01 | 12.11 | 3.42 |
| 620.96 | 172.37 | 24.72 | 17.53 | 17.35 | 4.86 | 18.10 | 14.39 | 4.26 |
| 732.32 | 185.79 | 21.03 | 4.15 | 8.62 | 0.58 | 17.98 | 14.97 | 3.76 |
| 741.13 | 239.07 | 27.94 | 6.35 | 13.27 | 0.92 | 19.20 | 15.92 | 4.07 |

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
