# Peer review of "Systematic Integration of Energy-Optimal Buildings With District Networks"

_energies, doi:10.3390/en12152945_

Round 1
Reviewer 1 Report
The topic of this paper is very interesting, specially during the climate change concern age. It proposes a double optimization approach with meta-models to link analysis and optimization at both building and urban scales, aiming at providing a systematical method to integrate multi-energy networks and low carbon resources in cities. The approach could be applied to design optimized energy networks in other regions which have similar environmental and economy concerns on the energy demands and supplies.
There are many energy related studies in urban planning and settlement, but most of them were either at an individual building level or at an urban level only. This paper applies the combination of those interacted systems together and studies the energy consumption, i.e., a fully electric energy providing system. This is a very interesting and important idea because it not only takes the economy into account, but also puts the environmental protection as a factor. The study found the “long-term power-to-gas storage” combining with “lower temperature CO2 networks” system can reduce energy (operating) cost and CO2 emission dramatically.
The paper has very good summary of energy supply and demand trend and challenges, and broadly references the current energy usage studies at different scales in urban development, However, those existing studies seems lack the interaction between different scales and levels, which are actually connected and impacted with each other, and this might lead to an incomplete picture of the urban energy consumption pattern. The “long-term power-to-gas storage system” indeed is efficient and sustainable technology, which includes PV panel, energy storage, and distributed energy resource, and I believe the “low temperature CO2 network” consumes the CO2 generated from the distributed energy resource, this network not only reduces the energy demand, but also controls CO2 emission and hence is environment favorable. The case study is carefully designed and planned and has made good comparison and contrast between two different time resolutions. The case study found the its method is economic and environment friendly to urban development. The results have been discussed in detail and are very interesting. In addition, the paper points out further study directions to enhance this method. The conclusion is generally supported by the study.
The paper states “a full electric energy providing system” leads reduction for both operating cost (for energy) and CO2 emission. This is the foundation of the study, in addition, because the majority of readers are the urban planners who might not have great energy and environment knowledge, so it would be better to have an explicitly explanation to the system. The paper concludes that “can lead to operating cost saving of 48% and CO2 emission saving up to 100%”, the study result indeed shows the energy (operational) cost and CO2 emission reduction but does not have the support to those numbers. Since this conclusion is significant and suggest the authors summarize the study results to draw this conclusion. In addition, there is no mention to the source of CO2, nor actual amount consumption of CO2 in the “system”, nor the time correlation between the CO2 generation and consumption, therefore recommend the authors have those explanations.
Reviewer 2 Report
This paper applies process integration techniques within a mixed integer linear programming formulation to evaluate optimal energy conversion technologies for different district energy networks and potential waste heat recovery from industrial plants using heat signature models and climate data to build a parameterized residential sector profile.
The work does not seem to be well organized and structured: first of all, although within the abstract section authors state to validate the model using Rotterdam, NL as a case study, within the paper no mention is given to Rotterdam city, but all calculations are performed to Geneva city (are authors sure of having sent the right abstract?).
Even neglecting this detail, which is very confusing for the reader, the paper lacks of a clear statement of its aim and usefulness, thus also the novelty, for the scientific community.
I therefore recommend this paper for Major Revisions.
In the following, I give specific suggestions and comments about the paper:
1. Introduction section
It lacks of clarity regarding paper’s aim, purpose, objectives and methodology applied. Although a brief mention is given in lines 67-72, it is in the reviewer’s opinion that the above-mentioned should be clearly specified in order that the reader understand the logical flow of the paper. In other words: why, what and how did you do in the article? See also lines 117-123, 137-145 which should be placed within the introduction section and not in methodology section.
2. Section 3.1
- What do authors mean with meta-models? Is it a validated expression/nomenclature in the scientific community? If not, please provide additional references.
- In which way did authors perform the grouping of buildings referred in lines 170-171 (i.e. basing on regulation about building energy performance/efficiency or on statistical analysis?)?
- Lines 171-176 are not clear.
3. Section 3.2
Mathematical formulation is hard to read, maybe a summary section with a flow chart (if applicable) could be useful for a better reading.
4. Case study section:
Lines 272-280: how did you account for behavioural aspects of energy demand?
5. Section 3.4
Figure 10: Periods from 1 to 8 in legend are typical days?
Equations 12 and 13: why didn’t you formulate this indexes in terms of demand?
6. Results and Discussion section:
Figure 13 is not clear, why bars are given in terms of investment cost (x-axis)?
Figure 17: for electricity, y-axis refers to import-export. Is it really imp/exp or demanded/produced?
For natural gas, for which reason the legend gives only NG import although the axis gives imp/exp (see also figure 19)?
7. Conclusions section
Conclusions seem to be a summary of results section.
Which is the added value of this work? Where do the novelty rely?
Round 2
Reviewer 2 Report
The authors reviewed the paper as required, thus improving the final outlook of the article.
For this reason, I believe that the paper should be accepted in its present form.